# YTHDF2 governs muscle size through a targeted modulation of proteostasis

Christopher J. Gilbert[1,2], Charles P. Rabolli[1,2], Volha A. Golubeva[1], Kristina M. Sattler ®[1], Meifang Wang[3,4], Arsh Ketabforoush ®[3,4], W. David Arnold[3,4,5,6,7,8], Christoph Lepper[1] & Federica Accornero ®[1,2] ✉

The regulation of proteostasis is fundamental for maintenance of muscle mass and function. Activation of the TGF-β pathway drives wasting and premature aging by favoring the proteasomal degradation of structural muscle proteins. Yet, how this critical post-translational mechanism is kept in check to preserve muscle health remains unclear. Here, we reveal the molecular link between the post-transcriptional regulation of m⁶A-modified mRNA and the modulation of SMAD-dependent TGF-β signaling. We show that the m⁶A-binding protein YTHDF2 is essential to determining postnatal muscle size. Indeed, muscle-specific genetic deletion of YTHDF2 impairs skeletal muscle growth and abrogates the response to hypertrophic stimuli. We report that YTHDF2 controls the mRNA stability of the ubiquitin ligase ASB2 with consequences on anti-growth gene program activation through SMAD3. Our study identifies a post-transcriptional to post-translational mechanism for the coordination of gene expression in muscle.

Skeletal muscle is essential to maintaining such functions as voluntary locomotion, thermostasis, metabolism, and postural maintenance[1–4]. Unlike other tissues, the growth of adult skeletal muscle is driven by an increase in the size (hypertrophy) of post-mitotic muscle fibers, rather than by cellular proliferation. Myofiber size decreases with aging, leading to a loss in muscle mass and quality, and predisposing individuals to physical impairment and heightened mortality risks[5–7]. Given that no consensus has been reached for pharmacologic intervention to prevent muscle wasting, lifestyle approaches are recommended to attenuate mass reduction, namely mechanical overload via resistance training[8–10]. However, many individuals fail to respond to anabolic stimuli or cannot partake in such activity[6,11]. Therefore, it is imperative that we gain greater insight into the factors that orchestrate muscle maintenance to identify therapeutic avenues for myopathies driven by muscle mass loss.

As protein levels in muscle correlate to mass acquisition, the regulation of proteostasis (protein turnover) is essential to healthful muscle maintenance. Previous studies point to a proteostatic imbalance favoring protein catabolism (decay) in aged muscle, leading to a blunted response to growth stimuli[12,13]. Catabolic signaling can be triggered by the transforming growth factor-β (TGF-β) pathway[14]. Within this, the activation of transcription factors SMAD2/3 is the primary driver of skeletal muscle wasting, whereby anti-hypertrophic (anti-growth) genes such as the ubiquitin ligases MAFbx/Atrogin-1 and Muscle RING Finger-1 (MuRF1) are upregulated[15–18]. Atrophic muscle is consequently characterized by enhanced protein degradation through the ubiquitin system[19–22]. Understanding how these signal networks intersect is essential to understanding how the proteome remodels during muscle wasting.

For proteostatic post-translational mechanisms to occur, gene expression must first take place. Beyond transcription, gene regulation

[1]Department of Physiology and Cell Biology, Dorothy M. Davis Heart and Lung Research Institute, The Ohio State University, Columbus, OH, USA. [2]Department of Molecular Biology, Cell Biology and Biochemistry, Brown University, Providence, RI, USA. [3]NextGen Precision Health, University of Missouri, Columbia, MO, USA. [4]Department of Physical Medicine and Rehabilitation, University of Missouri, Columbia, MO, USA. [5]Department of Neurology, University of Missouri, Columbia, MO, USA. [6]Department of Medical Pharmacology and Physiology, University of Missouri, Columbia, MO, USA. [7]Division of Neuromuscular Disorders, Department of Neurology, The Ohio State University, Columbus, OH, USA. [8]Department of Physical Medicine and Rehabilitation, The Ohio State University, Columbus, OH, USA. ✉e-mail: federica_accornero@brown.edu

undergoes extensive post-transcriptional control, and chemical modifications to mRNA can have far-reaching implications on transcript fate[23–25]. Our group identified methylation at the $N^6$ position of adenosine ($m^6A$) as a key determinant of adult muscle size[26]. $m^6A$ is catalyzed by methyltransferase-like 3 (METTL3) and is considered dynamic and reversible due to the activity of the demethylases fat mass- and obesity-associated protein (FTO) and AlkB Homolog 5 (ALKBH5)[27–30]. We found that myofiber-specific deletion of METTL3 elicits wasting and blunted hypertrophic growth, highlighting the significance of this modification for muscle homeostasis[26]. Yet, how $m^6A$-modified mRNAs are regulated, downstream of catalysis, in muscle remains unknown.

The YT521-B homology (YTH) domain-containing family of proteins (YTHDF) 1–3 specifically recognizes and binds $m^6A$-containing transcripts[31–33]. Several studies purport distinct roles for these proteins, designating YTHDF1 as a promoter of mRNA translation and YTHDF2 as an arbiter of transcript decay, while the role of YTHDF3 remains more obscure[33–37]. Others, however, suggest functional redundancy of these family members[38,39]. Understanding how these proteins function to integrate context-dependent signals in muscle may shed light on the mechanisms that underlie muscle proteome remodeling during growth or wasting.

Here, we report a role for $m^6A$-binding protein YTHDF in postnatal muscle growth and stress adaptation. Using genetically engineered mice with skeletal muscle-specific deletion of YTHDF2, we observe reduced muscle size and function with aging. We further demonstrate the necessity of YTHDF2 for the hypertrophic response to muscle overload. Mass spectrometry analysis identifies a YTHDF2-dependent regulation of the ubiquitin ligase ASB2 with downstream SMAD3 dysregulation to favor growth inhibition. Altogether, we have discovered an indispensable role for YTHDF2 in skeletal muscle, revealing a mechanism for the regulation of proteostasis and resultant muscle mass control.

## Results

### Loss of YTHDF2 limits adult skeletal muscle size
The regulation of $m^6A$ content has been linked to the maintenance of muscle homeostasis[26]. To determine how the effects of this mRNA modification are implemented, we generated skeletal muscle-specific knockout mice for the $m^6A$-binding protein YTHDF2 (Y2-KO) (Fig. 1a–d). Skeletal muscle specificity was conferred through myogenin (Myog)-driven Cre system[40] (Fig. 1a), which was effective in deleting YTHDF2, as shown both at the mRNA (Fig. 1b) and protein level (Fig. 1c, d) from total muscle extracts. Mice lacking YTHDF2 in muscle were indistinguishable from their control littermates at weaning, but muscle growth occurring between 1 and 2 months of age was defective in Y2-KO animals (Fig. 1e–g). This abnormality was independent of mouse size, as body weight and tibia length were unaffected (Fig. 1h, i), and histopathological analysis showed no overall disruption of muscle architecture (Fig. 1j). We detected no global change in $m^6A$ content, excluding contribution of aberrant modification to the observed growth impairment (Fig. 1k); moreover, replenishment of YTHDF2 via adeno-associated virus (AAV) was sufficient to rescue Y2-KO muscle size defects (Fig. 1l, m) without impacting wild-type muscle size (Supplementary Fig. 1). To understand what drove the observed reduction in muscle mass in the absence of YTHDF2, we analyzed myofiber cross-sectional areas and detected reduced size of Y2-KO myofibers compared to control littermates at 2 months (Fig. 2a–d). Interestingly, fiber-type distribution was unchanged with loss of YTHDF2 (Fig. 2e, f), suggesting that regulation of myofiber size, and consequent muscle mass, by YTHDF2 showed no bias toward fiber-type specification or differentiation.

At 2 months of age, not all analyzed skeletal muscles were affected by the loss of YTHDF (Supplementary Fig. 2), which led us to question whether a more global phenotype could develop with aging. Indeed, 8-month-old Y2-KO mice showed extensive reduction

in the mass of all analyzed hindlimb muscles (Fig. 3a–e) without appreciable effects on heart size (a control organ not targeted by the used genetic deletion system) or body size, as indicated by tibia length (Fig. 3f, g). Echoing muscle mass, correspondent myofiber areas were all smaller in 8-month-old Y2-KO mice (Fig. 3h–m). To determine if the observed change in size led to functional deficits, we subjected mice to in vivo muscle contractility measurements and exercise testing. We detected reduced tetanic muscle torque production (Fig. 3n) without changes in fatigability in Y2-KO animals (Supplementary Fig. 3a). Further, running performance was impaired in mice with muscle YTHDF2 deficiency (Fig. 3o, p). These data demonstrate the necessity of YTHDF2 for postnatal muscle size determination and function.

To better understand the molecular signature accompanying Y2-KO muscle defect, we first tested for metabolic components such as glycogen concentration and succinate dehydrogenase (SDH) activity, where we observed non-significant changes (Supplementary Fig. 3b, c). We then assessed the levels of key anabolic and catabolic factors (Fig. 4a–o). While we detected no significant disruption in protein synthesis markers (phosphor-)AKT or mTOR (Fig. 4a–g), autophagic markers LC3II and Beclin-1 were upregulated (Fig. 4i–l). We also observed increased polyubiquitination in Y2-KO muscles, despite no change in the abundance of monomeric ubiquitin (Fig. 4m, n). Further, we detected an increase in proteasomal activity of Y2-KO muscles (Fig. 4o). These results suggest proteostasis is overall altered through enhancement of protein catabolism in muscles deficient for YTHDF2.

### YTHDF2 is necessary for skeletal muscle overload-induced hypertrophy and participates in reparative muscle growth
Muscle growth is essential to achieve full functionality in adulthood and reparative capacity following injury or insult. To test the role of YTHDF2 in muscle regeneration, we employed an established chemical injury model through barium chloride injection of tibialis anterior muscle[41]. This insult elicits muscle necrosis (first ~4 days) followed by myofiber formation (typically completed by day 7); the latter requires activation of myogenesis to reform muscle and is followed by hypertrophic growth of newly generated myofibers[42] (Fig. 5a). Injured muscles from Y2-KO animals failed to reach control size, halting their growth to match the observed reduction in mass at baseline (Fig. 5b). Interestingly, knockout muscles fared comparably to controls up to 7 days following injury (Fig. 5b–d), suggesting that YTHDF2 controls muscle size independent from myofiber formation in our model. To further corroborate these data, we determined the fusion index of isolated control and Y2-KO myoblasts after 4 days of differentiation and saw no effect on myoblast fusion in the absence of YTHDF2 (Fig. 5e, f). In line with baseline findings at 8 months, molecular assessment of Y2-KO muscles 7 days post-injury shows enhanced protein polyubiquitination with no change in AKT activation (Supplementary Fig. 4). This injury model also highlighted higher mTOR levels (Supplementary Fig. 4e, f) perhaps suggesting attempted compensation by Y2-KO muscles to promote growth, and an even more significant elevation of LC3 and p62 levels (Supplementary Fig. 4h–k). Altogether, these data highlight proteostasis perturbation in injured YTHDF2-deficient muscles and suggest their growth defect is independent of myoblast fusion.

The ability of muscle to grow is not only important for postnatal maturation or regeneration; it is an essential process to counteract increases in workload through hypertrophy, achieved by increasing myofiber size above the baseline level. To test if YTHDF2 participates in hypertrophic remodeling, we first subjected wild-type mice to plantaris overload stress, where we detected an increase in YTHDF2 expression (Fig. 6a–c), suggesting the responsiveness of this protein to hypertrophic stimuli. We then performed overload surgeries on control and Y2-KO at 2 months of age. This timepoint allowed us to avoid

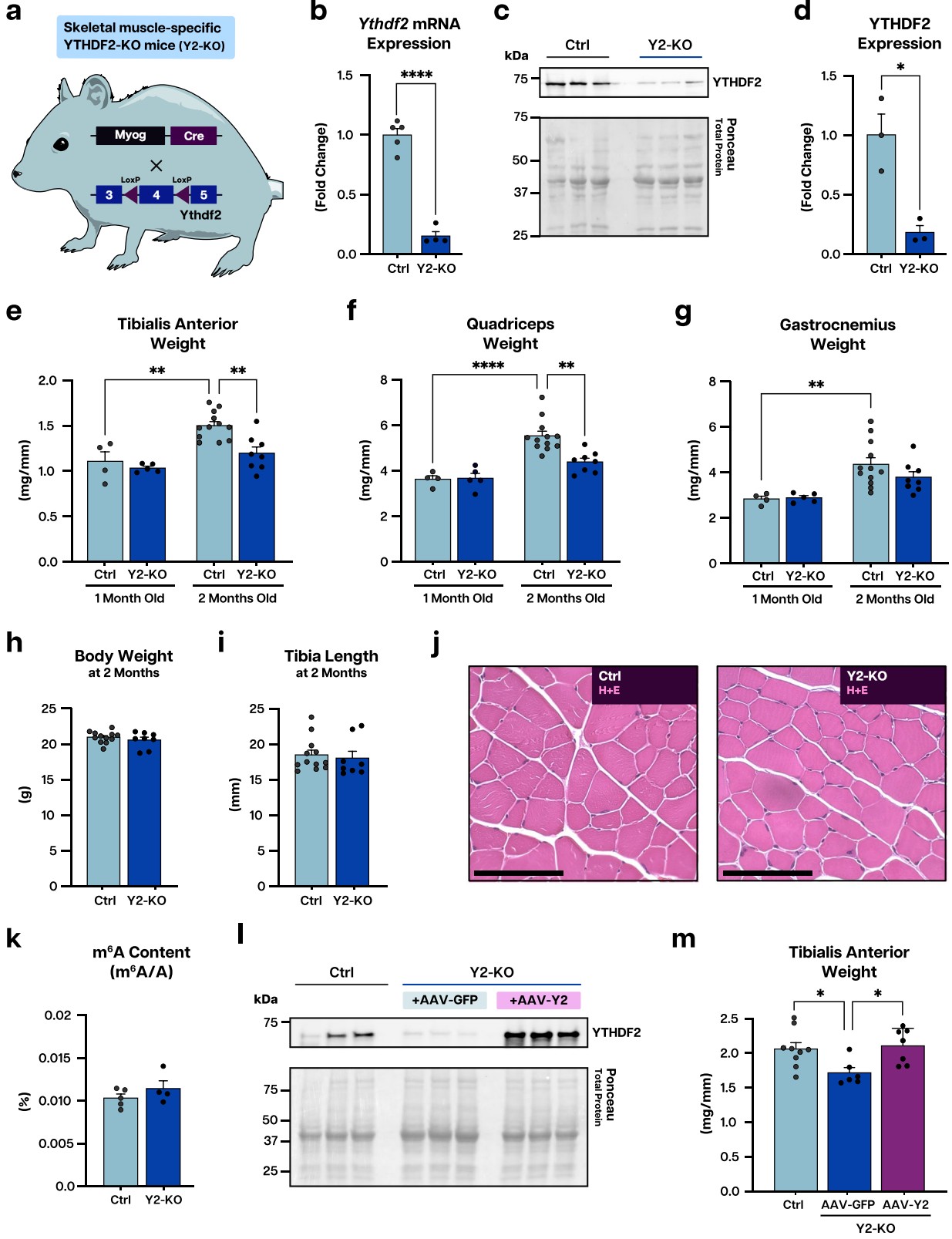

the contribution of preexisting size deficits to our findings, as it preceded size divergence between Y2-KO and control plantaris muscles (Supplementary Fig. 2b, e). Remarkably, the response to overload-induced hypertrophy was completely abrogated in Y2-KO mice, both at the level of muscle mass (Fig. 6d) and myofiber area (Fig. 6e, f), illustrating the requirement of YTHDF2 for the hypertrophic response of muscle to overload stress.

## YTHDF2 regulates the fate of m⁶A-mRNA encoding ASB2

To understand the mechanism behind YTHDF2-mediated size regulation, we assessed the proteomic reprogramming of muscle undergoing hypertrophy. We performed our molecular analysis following 2 days of overload stress to capture the initial changes that drive muscle remodeling without confounding effects derived from overt size differences in Y2-KO samples. Using mass spectrometry, we identified a

**Fig. 1 | YTHDF2 regulates postnatal skeletal muscle mass. a** Schematic of muscle-specific YTHDF2 knockout (Y2-KO) mouse generation, created in Adobe Illustrator. **b** qPCR analysis of *Ythdf2* mRNA expression using the gastrocnemius of control (Ctrl) and Y2-KO mice. **c** Western blot analysis of YTHDF2 protein expression in Ctrl and Y2-KO gastrocnemius. **d** Quantification of YTHDF2 protein expression using total protein detection by Ponceau stain as a loading control. **e** Tibialis anterior weight, **f** quadriceps weight, and **g** gastrocnemius weight for 1- and 2-month-old Ctrl and Y2-KO mice, normalized to tibia length. **h** Body weight and **i** tibia length for Ctrl and Y2-KO mice at 2 months of age. **j** Representative hematoxylin/eosin (H&E) staining for Ctrl and Y2-KO tibialis anterior. **k** Quantification of m⁶A relative to total adenosine (m⁶A/A) as determined by ELISA in Ctrl and Y2-KO quadriceps. **l** Western blot analysis of YTHDF2 protein expression in Ctrl tibialis anterior compared to Y2-KO injected with AAV-GFP or AAV-YTHDF2 (AAV-Y2) at 2 months of age and analyzed 8 weeks following administration. **m** Tibialis anterior weight normalized to tibia length for the indicated groups. Biological animal replicates: $n = 5$ (Ctrl) and 4 (Y2-KO) in panel **b**; $n = 3$ (Ctrl) and 3 (Y2-KO) in panel **c** and **d**; $n = 4$ (Ctrl at 1 month), 5 (Y2-KO at 1 month), 12 (Ctrl at 2 months), and 8 (Y2-KO at 2 months) for panel **e**–**g**; $n = 12$ (Ctrl) and 8 (Y2-KO) in panel **h, i**; $n = 5$ (Ctrl) and 4 (Y2-KO) in panel **k**; $n = 3$ (Ctrl), 3 (Y2-KO + AAV-GFP), and 3 (Y2-KO + AAV-Y2) in panel **l**; $n = 9$ (Ctrl), 6 (Y2-KO + AAV-GFP), and 7 (Y2-KO + AAV-Y2) in panel **m**. Data were presented as the mean ± SEM with the individual biological samples shown. Significance was determined by two-tailed Student's *t*-test for comparisons between Ctrl and Y2-KO mice, by two-way ANOVA with Tukey's HSD multiple-comparison test for comparison of the means of Ctrl and Y2-KO mice at 1 and 2 months of age, or by one-way ANOVA for Ctrl, Y2-KO + AAV-GFP, and Y2-KO + AAV-Y2 mice: *$p ≤ 0.05$, **$p ≤ 0.01$, ****$p ≤ 0.0001$. Scale bar = 100 μm for panel j.

total of 2624 proteins (peptides ≥3), of which 2.13% (32) were differentially regulated by YTHDF2 with at least 2-fold change (Fig. 6g and Supplementary Data 1). Given the postulated role of YTHDF2 in targeted m⁶A-mRNA decay, we focused on the proteins upregulated in its absence. Our analysis showed ASB2 as the most upregulated protein in Y2-KO samples (Fig. 6h). ASB2 (Ankyrin repeat and SOCS Box containing 2) is an E3 ubiquitin ligase complex component that has been implicated in growth inhibition in muscle and beyond[21,43–46]. In line with our phenotypic observations, ASB2 overexpression is sufficient to induce muscle atrophy[21,43]. To gain additional insights into the impact of YTHDF2 in skeletal muscle gene regulation, we performed YTHDF2 RNA immunoprecipitation and sequencing (RIP-Seq) and compared the detected transcripts with those previously found with m⁶A (me-) RIP-Seq (Fig. 6i and Supplementary Data 2). Of these overlapping transcripts, we filtered according to whether the corresponding gene product was detected via mass spectrometry and found that 26.7% of the detected proteins were regulated by YTHDF2 (Fig. 6j). *Asb2* was detected across all omics assessments, suggesting a pivotal role for the m⁶A-YTHDF2 axis in regulating ASB2 expression.

Considering the link between ASB2 and negative mass regulation, we sought to analyze how YTHDF2 alters the expression of this target in more detail. We found that loss of YTHDF2 increases the protein level of ASB2 in baseline muscle by nearly 10-fold over control by Western blot (Fig. 7a, b). Enhanced ASB2 protein expression was also confirmed at 8 months of age (Supplementary Fig. 5a, b) and following acute injury (Supplementary Fig. 5c, d). Contrarily, total *Asb2* mRNA was not proportionally increased (Fig. 7c), excluding a primary contribution of transcription to the observed ASB2 protein abundance. We did, however, detect increased *Asb2* mRNA in the cytosolic compartment of Y2-KO muscles, further implicating YTHDF2 in regulation of these transcripts following their nuclear export (Fig. 7d). Considering the post-transcriptional role for YTHDF2 on effecting m⁶A-modified mRNA, we assessed the methylation status of *Asb2* transcripts and confirmed that *Asb2* mRNA is m⁶A-modified in muscle (Fig. 7e). Importantly, RNA immunoprecipitation revealed binding of YTHDF2 to *Asb2* mRNA (Fig. 7f), suggesting a direct effect of YTHDF2 on ASB2 expression, aligning with our sequencing results. To test if the observed increase in ASB2 upon loss of YTHDF2 is due to changes in transcript half-life, we then performed an mRNA decay assay after blocking transcription with actinomycin D. We found that knocking down YTHDF2 significantly stabilized *Asb2* mRNA (Fig. 7g). These data show that YTHDF2 binds m⁶A-modified *Asb2* mRNA to favors its decay and, consequently, the loss of YTHDF2 increases ASB2 expression.

Considering the stark induction of ASB2 with Y2-KO, we then tested whether ASB2 downregulation in these muscles would be sufficient to rescue their response to hypertrophic stimuli. To address this, siRNA for ASB2 (si-ASB2) and a non-targeting control (si-Ctrl) were electroporated into Y2-KO plantaris immediately following muscle overload surgery (Fig. 7h). For Y2-KO mice treated with si-ASB2, post-overload muscle size paralleled that of control littermates, inextricably linking YTHDF2's role in muscle size determination to the regulation of ASB2 (Fig. 7i).

## Regulation of ASB2 by YTHDF2 perturbs SMAD3-dependent proteostatic programs

ASB2 has been linked to the TGF-β/SMAD pathway, a key axis to muscle growth and homeostasis[45]. To better understand the molecular events underpinning YTHDF2-ASB2 mediated mass regulation, we assessed the consequences of YTHDF2 deletion on the SMAD family of proteins in muscle. We found a striking upregulation of SMAD3, and to a lesser extent SMAD2, in Y2-KO muscle (Fig. 8a; quantification in Supplementary Fig. 6). SMAD3 is a transcription factor that activates anti-hypertrophic gene programming[15–18]. As a consequence of enhanced SMAD3 expression, we also observed an increase in active phosphorylated SMAD3 and a concomitant rise in DNA-binding cofactor FOXO3a (Fig. 8b–d), which has been reported to increase with SMAD3 augmentation[18]. Transcriptional targets *MAFbx/Atrogin-1* (Muscle-atrophy F-box protein) and *MuRF1* (muscle-specific ring finger protein 1) were increased in YTHDF2-deficient muscles (Fig. 8e, f) and chromatin immunoprecipitation analysis detected increased binding of SMAD3 to their promoters (Fig. 8g, h).

SMAD3 protein levels are post-translationally controlled by several ubiquitin ligases, including members of the SMURF (SMAD ubiquitin regulatory factor) family[47,48]. Of note, ASB2 not only regulates protein stability directly[49], but its upregulation can have broad consequences on the proteome through regulation of additional components of the ubiquitin-proteasome system[43]. In Y2-KO muscles, we detected a reduction in SMURF2 levels (Fig. 8i, j). Moreover, assessing protein stability with the translation inhibitor cycloheximide revealed a significant reduction in SMURF2 half-life in the absence of YTHDF2 (Fig. 8k, l). Importantly, the reduction in SMURF2 level by loss of YTHDF2 can be rescued by targeting ASB2 (Fig. 8m, n), confirming these factors work within the same signaling axis. This is further supported by our finding that SMURF2 stability is enhanced in the absence of ASB2 (Supplementary Fig. 7). Taken together, we report a role for YTHDF2 in controlling TGF-β-induced anti-growth signaling through ASB2 (Fig. 8o).

## Discussion

Skeletal muscle growth and maintenance require the coordinated regulation of protein synthesis and stability. Over the past decade, global profiling studies have revealed roles for m⁶A RNA modification in myoblast proliferation[50–55]. However, our understanding of m⁶A's role in postnatal muscle growth and homeostasis have remained nascent. While previous work from our group has shown the METTL3-m⁶A axis facilitates hypertrophy in vivo[26], the current study addresses how m⁶A elicits cellular consequences downstream of its catalysis in skeletal muscle. Here, we highlight a pivotal role for the m⁶A-binding protein YTHDF2 in governing muscle mass through restricted ubiquitin ligase ASB2 expression, overall modulating the ubiquitin ligase network to favor growth.

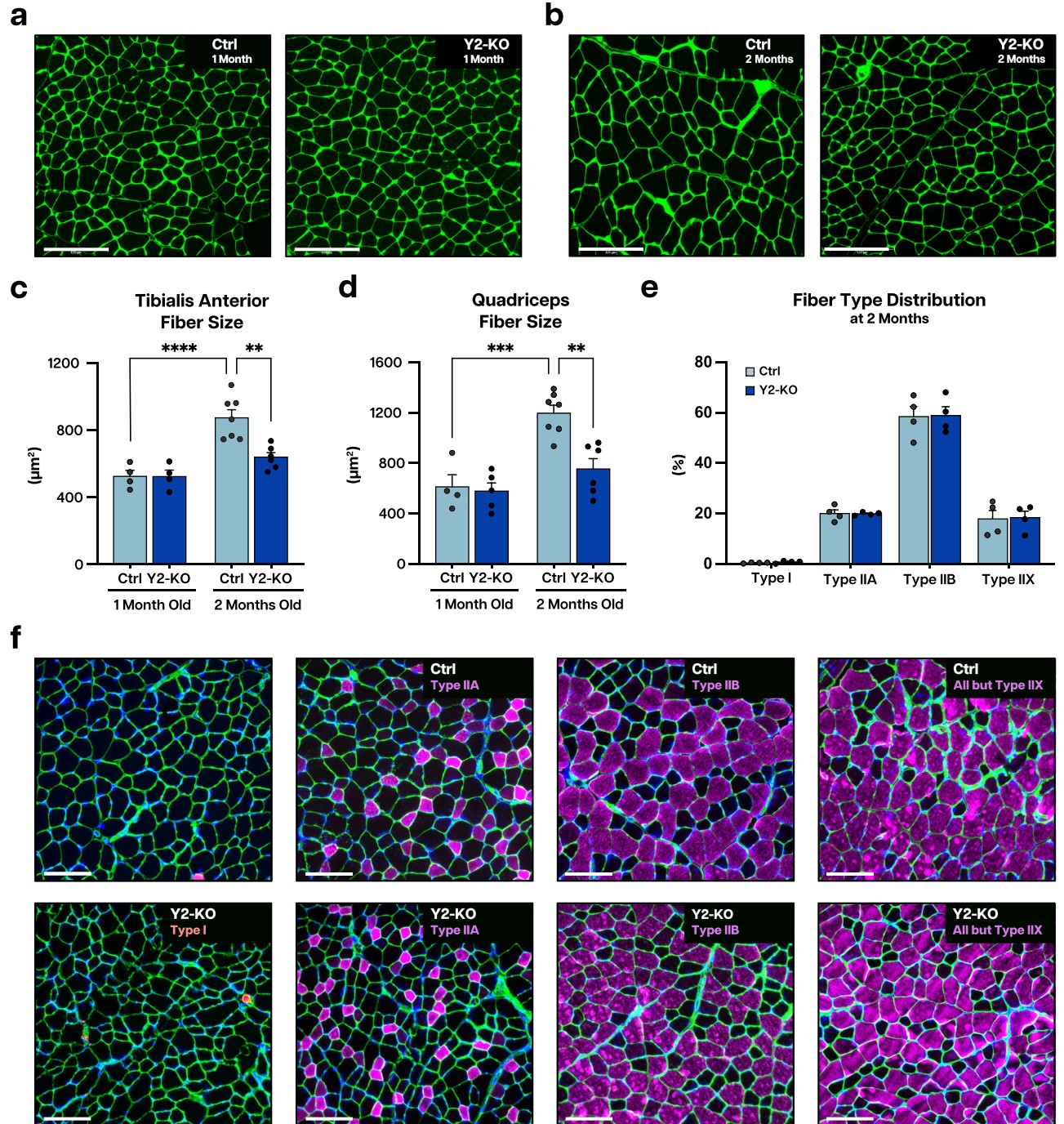

**Fig. 2 | Loss of YTHDF2 impairs myofiber growth irrespective of fiber-type distribution.** Representative wheat germ agglutinin (WGA, green) stained images for Ctrl and Y2-KO tibialis anterior at **a** 1 and **b** 2 months of age. Myofiber cross-sectional areas for **c** tibialis anterior and **d** quadriceps for Ctrl and Y2-KO mice at 1 and 2 months of age. **e** Fiber type distribution for Ctrl and Y2-KO tibialis anterior and **f** representative myosin type staining for type I (anti-BA-D5, red), type IIA (anti-SC-71, pink), type IIB (anti-BF-F3, pink), and all but IIX (anti-BF-35, pink) with counterstains anti-laminin (green) and DAPI (blue). Biological animal replicates $n = 4$ (Ctrl at 1 month), 4 (Y2-KO at 1 month), 7 (Ctrl at 2 months), and 6 (Y2-KO at 2 months) for panel **c, d**; $n = 4$ (Ctrl) and 4 (Y2-KO) for panel **e**. Data were presented as the mean ± SEM with the individual biological samples shown. Significance was determined by two-tailed Student's *t*-test for comparisons between Ctrl and Y2-KO mice at 2 months, or by two-way ANOVA with Tukey's HSD multiple-comparison test for comparison of the means of Ctrl and Y2-KO mice at 1 and 2 months of age. Welch's correction was used for unequal variances: $**p \le 0.01$, $***p \le 0.001$, $****p \le 0.0001$. Scale bar = 125 μm for panel **a, b**. Scale bar = 100 μm for panel **f**.

We report that muscles lacking YTHDF2 exhibit blunted growth starting from 2 months of age. Considering that postnatal myoblast proliferation and fusion peaks prior to 1 month of age[56], at which we detected no disparity between control and Y2-KO muscles, we can preclude the primary roles of YTHDF2 in developmental myofiber formation in our model. The defects in muscle mass at the 2-month time-point likely indicate the inability of muscles lacking YTHDF2 to undergo hypertrophic growth. The central role of YTHDF2 in the regulation of adult muscle size is further supported by our results showing a reduction in myofiber area in all hindlimb muscles of 8-month-old mice.

The failure of knockout muscles to respond to hypertrophic stress further hints toward the indispensability of YTHDF2 in muscle growth signaling. Interestingly, current disputes question whether YTHDF1-3 holds distinct regulatory roles or exhibits functional

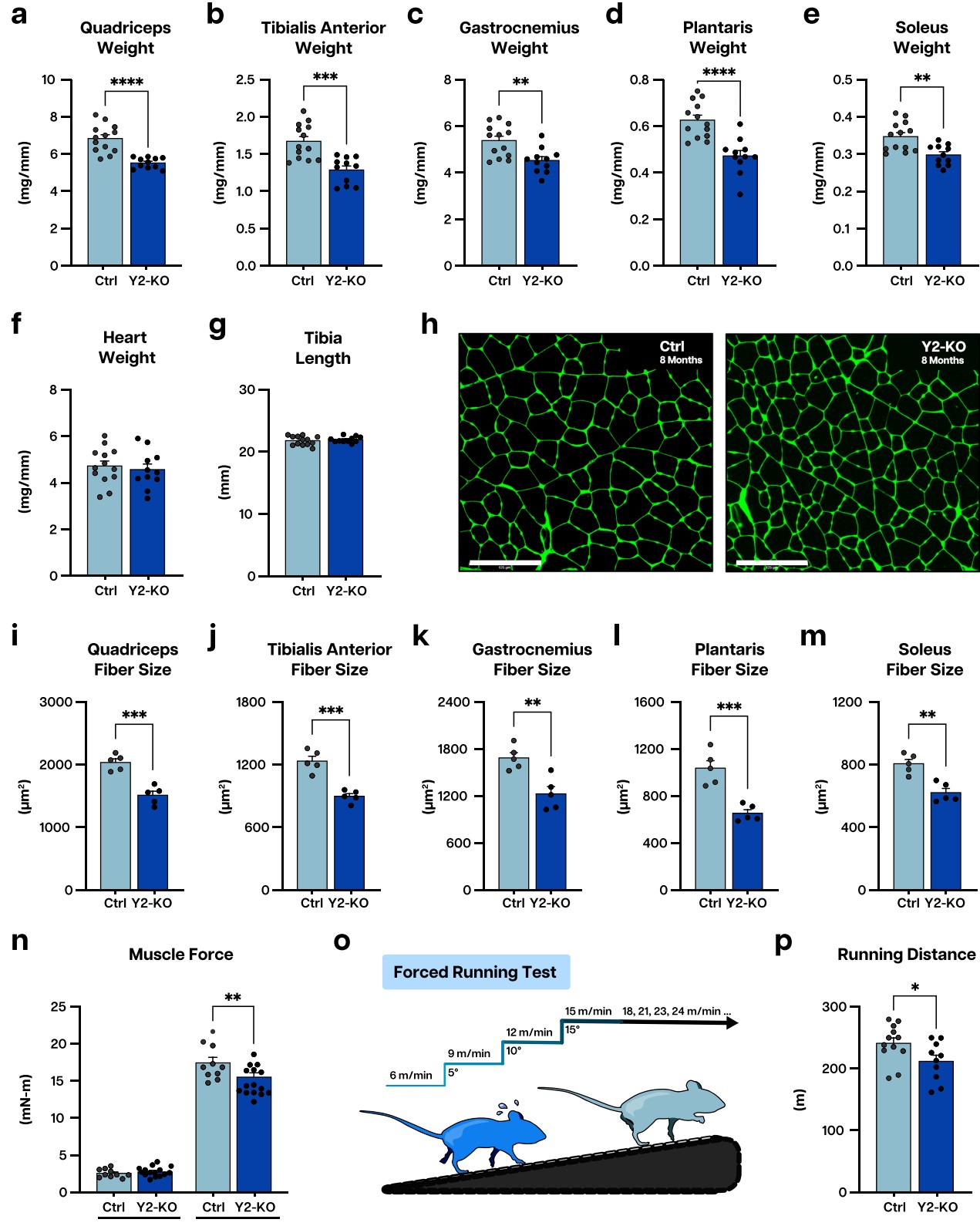

redundancy[33–36,38,39]. Aligning with the former assertion, our findings suggest the necessity of YTHDF2 for muscle homeostasis. Indeed, the failure of muscle to respond to growth stimuli without YTHDF2 shows an inability of other m6A-binding proteins to act as its proxy. In line with this, we recently observed a distinct functionality for YTHDF2 in the heart[57], and we now report such specificity in the context of postnatal skeletal muscle maintenance.

Mass spectrometry analysis of YTHDF2-deficient muscle revealed a stark upregulation of ASB2, a ubiquitin ligase previously connected to age-related muscle wasting[21,43–46]. Aberrant ubiquitin ligase expression elicits changes in proteostasis, triggering muscle wasting and impaired growth responses[12,13]. ASB2 has been noted to modulate the expression of other ubiquitin ligases in skeletal muscle and has been previously linked to SMAD regulation[20,43,45]. In the absence of YTHDF2,

**Fig. 3 | Deletion of YTHDF2 elicits disparity in muscle size and function by 8 months of age. a** Quadriceps weight, **b** tibialis anterior weight, **c** gastrocnemius weight, **d** plantaris weight, **e** soleus weight, **f** and heart weight for Ctrl and Y2-KO mice at 8 months of age, normalized to tibia length. **g** Tibia length for Ctrl and Y2-KO mice at 8 months of age. **h** Representative wheat germ agglutinin (WGA, green) stained images for Ctrl and Y2-KO tibialis anterior at 8 months of age. Myofiber cross-sectional areas for **i** quadriceps, **j** tibialis anterior, **k** gastrocnemius, **l** plantaris, and **m** soleus for Ctrl and Y2-KO mice at 8 months of age. **n** In vivo muscle twitch and tetanic torque measures for Ctrl and Y2-KO mice at 8 months of age. **o** Schematic for forced graded maximal exercise test, created in Adobe Illustrator, and **p** maximal running distance for Ctrl and Y2-KO mice at 8 months of age. Biological animal replicates: $n = 13$ (Ctrl) and 11 (Y2-KO) in panel **a–g**; $n = 5$ (Ctrl) and 5 (Y2-KO) in panel **i–m**; $n = 10$ (Ctrl) and 15 (Y2-KO) in panel **n**; $n = 12$ (Ctrl) and 10 (Y2-KO) in panel **p**. Data were presented as the mean ± SEM with the individual biological samples shown. Significance was determined by a two-tailed Student's *t*-test for comparisons between Ctrl and Y2-KO mice. Welch's correction was used for unequal variances. Grubbs' (ESD) tests were run using GraphPad Prism, and outliers were removed from analysis when applicable: $*p \leq 0.05$, $**p \leq 0.01$, $***p \leq 0.001$, $****p \leq 0.0001$. Scale bar = 125 μm for panel **h**.

we noted higher levels of anti-hypertrophic SMAD3 and enhanced transcription of downstream ubiquitin ligases MuRF1 and MAFbx/Atrogin-1[15–18]. The upregulation of SMAD3 can, at least in part, be explained through the downregulation of SMURF2, another ubiquitin ligase that inhibits TGF-β-induced anti-hypertrophic signaling[47,48,58]. These data support our previous assertion that the m6A pathway is connected to TGF-β signaling in adult muscle[26].

In identifying an interaction between ASB2 and SMURF2, we propose that YTHDF2 functions to inhibit catabolic signaling through modulation of ASB2 levels. We have shown that YTHDF2 binds m6A-Asb2 transcripts to facilitate their decay, explaining the inverse relation between ASB2 and YTHDF2 expression and consequent impacts on anti-hypertrophic SMAD signaling. Previous reports have also linked the m6A demethylase FTO to ASB2 suppression[59], which suggests harmonization of the m6A network in cementing pro-growth cascades. It should be noted that the roles played by other m6A-binding proteins remain uncertain in the context of skeletal growth responses, leaving room for future assessments. Despite the essentiality of YTHDF2 in homeostasis, our data do not exclude roles for YTHDF1 and YTHDF3 in muscular processes, nor are our findings exhaustive of YTHDF2's contributions to muscle maintenance. For example, while our in vivo rescue result clearly shows the necessity of ASB2 for the abrogated hypertrophic response observed in YTHDF2-deficient muscles, we cannot currently exclude a role for other contributing ASB2 targets other than SMURF2, or even other proteins stabilized by loss of SMURF2 in addition to SMAD3. Beyond ASB2, our bioinformatic analyses revealed altered protein expression of several YTHDF2-bound transcripts in YTHDF2-deficient muscles. In line with the observed muscle mass reduction, we saw an increase in FBXO40, which has been shown to inhibit insulin-like growth factor 1-induced hypertrophy[60]. We detected the downregulation of molecular chaperones ANP32E and TOR1b, suggesting roles for YTHDF2 in regulating chromatin remodeling and protein folding events[61,62]. Additionally, upregulation of CAB39 (MO25), MAPKAP3, and SYNPO2L suggest dysregulated signal transduction in the absence of YTHDF2[63–66]. Further, while we detected no significant shift in glycogen concentration or SDH activity in muscles lacking YTHDF2, our findings do not preclude the contribution of other metabolic cascades to our phenotype. Indeed, previous reports have identified a role for ASB2, in fatty acid β-oxidation[43]. Our detection of enhanced PLIN4 expression with YTHDF2 deletion may also hint at a role for this axis in lipid storage[67].

While we recognize the complexity of the network for proteostatic regulation elucidated by the presented work, we can conclude that our study identifies a YTHDF2-dependent mechanism of skeletal muscle size regulation, connecting a post-transcriptional gene regulatory mechanism to post-translational control of protein stability, and rousing potential to address muscle wasting pathologies.

## Methods
### Ethics declarations
All presented experiments comply with the standards set forth by the Institutional Animal Care and Use Committee at The Ohio State University, and the Guide and Care and Use of Laboratory Animals published by the US National Institute of Health. All procedures are approved by The Ohio State University Institutional Animal Care and Use Committee and Institutional Biosafety Committee under protocol 2015A00000115-R2.

### Animal generation
Male and female C57BL6/N mice up to 8 months of age were used in this study. Mice were housed at 72 °F (22 °C) at 50% humidity under a 12-h light/ 12-h dark cycle with ad libitum access to a standard chow diet and water. *Ythdf2* LoxP-targeted (flox; fl) mice (*Ythdf2*fl/fl) were generated by Cyagen (Santa Clara, CA, USA), targeting exon 4. *Ythdf2*fl/fl mice were crossed with mice expressing the Cre recombinase gene under the control of the skeletal muscle-specific Myogenin (MyoG) promoter[40] to obtain muscle-restricted deletion of *Ythdf2* (Y2-KO). *Ythdf2*fl/fl littermates not expressing Cre recombinase were used as controls. Genotypes were cohoused, and mice were randomly allocated to study groups based on the order they were tagged for identification.

### Animal procedures and treatments
All experimental procedures were initiated on mice between 1 and 8 months of age. All operations were performed under 2–3% vaporized isoflurane. The assessment of myogenic repair was accomplished by inducing acute injury by 50-μL intramuscular injection of 1.2% BaCl2 in PBS to the tibialis anterior. PBS was injected contralaterally for use as an internal control.

Hypertrophic overload of the plantaris was achieved through bilateral synergistic ablation of the soleus and gastrocnemius. Lateral incisions were made on the bilateral lower hindlimbs, allowing for the exposure and subsequent removal of the distal and proximal tendons of the soleus and the distal tendon and proximal gastrocnemius head as described[68]. Mice were treated with Ethiqa XR (Covetrus, Dublin, OH, USA) 72-h extended-release buprenorphine prior to operation, and water was supplemented with ibuprofen 24 h pre- to 96 h post-operation. Tissues were collected after 2 weeks.

For siRNA-meditated rescue, 30 μL of 100 μM control non-targeting siRNA (IDT, #51-01-14-04), or si-ASB2 TriFECTa DsiRNA Kit (IDT, rn.Ri.Asb2.1–3) was electroporated into the plantaris muscle of Y2-KO mice immediately following synergistic ablation. Mice were anesthetized and the lower hindlimb was injected with 30 μl of 2 mg/ml Hyaluronidase per limb (#P4D14907, Worthington) with an insulin syringe (#309625, BD). One-hour post-recovery, mice were re-anesthetized, and surgery was performed. siRNA solution was administered to the plantaris muscle (about the surgical pocket) prior to suturing, after which two sets of electrical pulses were applied (-180 V). Animals returned to function movement within 15–30 min and tissues were harvested at an experimental endpoint, 2 weeks following surgery.

Adeno-associated viral overexpression of YTHDF2 was achieved using AAV9 vectors produced by Vector Biolabs (Malvern, PA, USA). Neonatal mice were injected intraperitoneally with $1 \times 10^{12}$ viral genome particles of AAV9-YTHDF2 (AAV9-CMV-m-Ythdf2) or control vector (AAV9-CMV-eGFP) and analyzed 8 weeks later. For adult recovery studies, the tibialis anterior of 2-month-old mice was injected with $1 \times 10^{12}$ viral genome particles of AAV9-YTHDF2 or control vector and analyzed 8 weeks following injection.

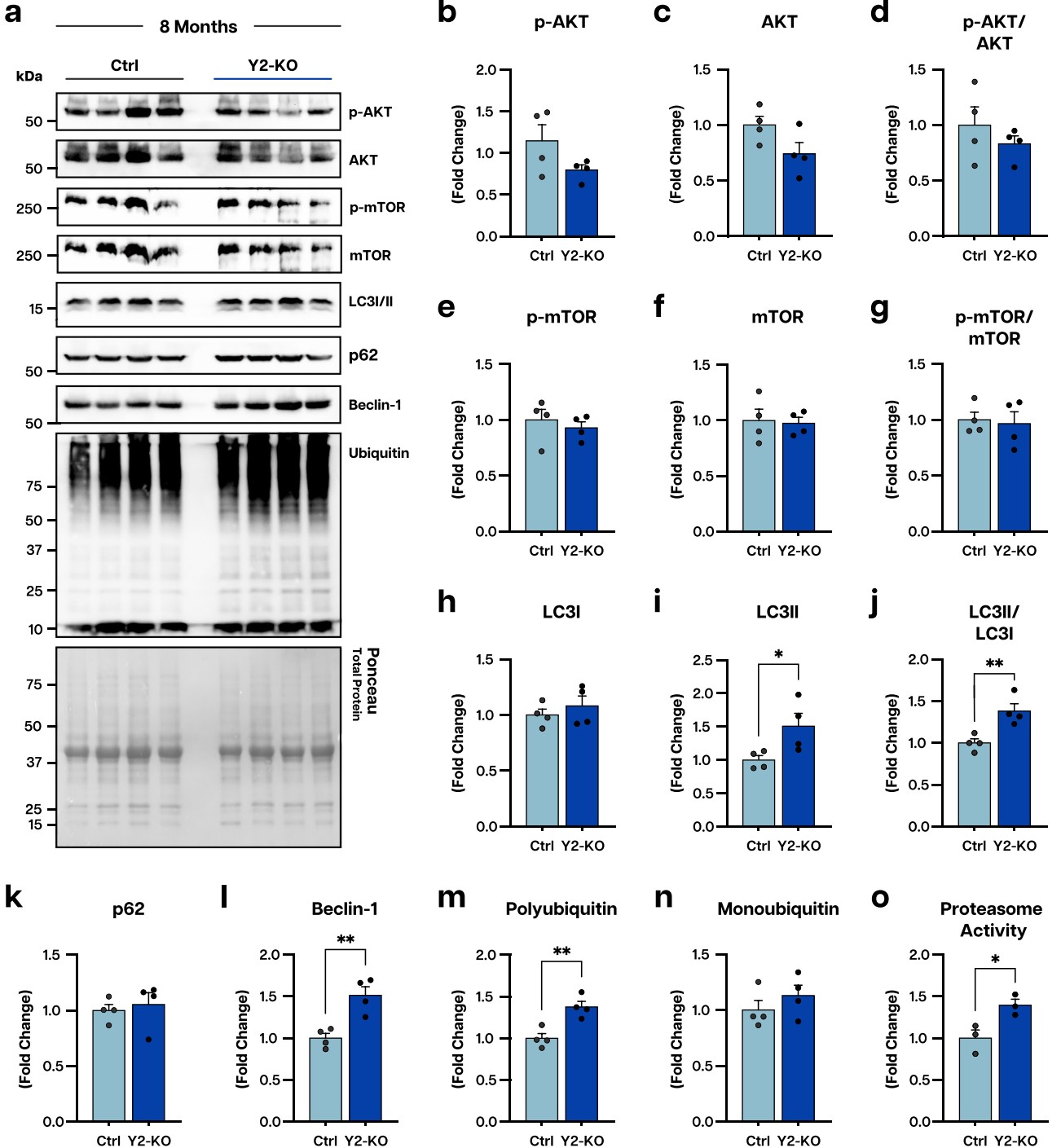

**Fig. 4 | YTHDF2-null muscles favor catabolic signaling. a** Western blot analyses and quantification using total protein detection by Ponceau stain as a loading control for **b** phosphor-(p-)AKT, **c** AKT, **d** p-AKT/AKT ratio, **e** p-mTOR, **f** mTOR, **g** p-mTOR/AKT ratio, **h** LC3I, **i** LC3II, **j** LC3II/LC3I ratio, **k** p62, **l** Beclin-1, **m** monoubiquitin, and **n** polyubiquitin in Ctrl and Y2-KO quadriceps at 8 months of age. **o** Proteasomal activity as determined via colorimetric proteasome activity assay kit in Ctrl and Y2-KO quadriceps. Biological animal replicates: $n = 4$ (Ctrl) and 4 (Y2-KO) in panel **a**–**n**; $n = 3$(Ctrl) and 3 (Y2-KO) in panel **o**. Data were presented as the mean ± SEM with the individual biological samples shown. Significance was determined by a two-tailed Student's $t$-test for comparisons between Ctrl and Y2-KO mice. Welch's correction was used for unequal variances: *$p \le 0.05$, **$p \le 0.01$.

Animal use was approved by the Institutional Animal Care and Use Committee at The Ohio State University.

**Exercise treadmill testing**

Exercise tests were conducted via a comprehensive lab animal monitoring system (CLAMS) with data collection using OxyMax software Ver 2.4.2 (Columbus Instruments, Columbus, OH, USA) using

previously described methods[69]. Following a period of acclimation, mice were subjected to an endurance or graded maximal exercise test. In brief, mice were placed on an enclosed treadmill at 0° incline, and the shock grid was activated. The treadmill speed (meters/minute), duration (minutes), and grade (degrees) were then increased until exhaustion was reached with the following parameters: 0 m/min, 3 min, 0°; 6 m/min, 2 min, 0°; 9 m/min, 2 min, 5°; 12 m/min, 2 min, 10°;

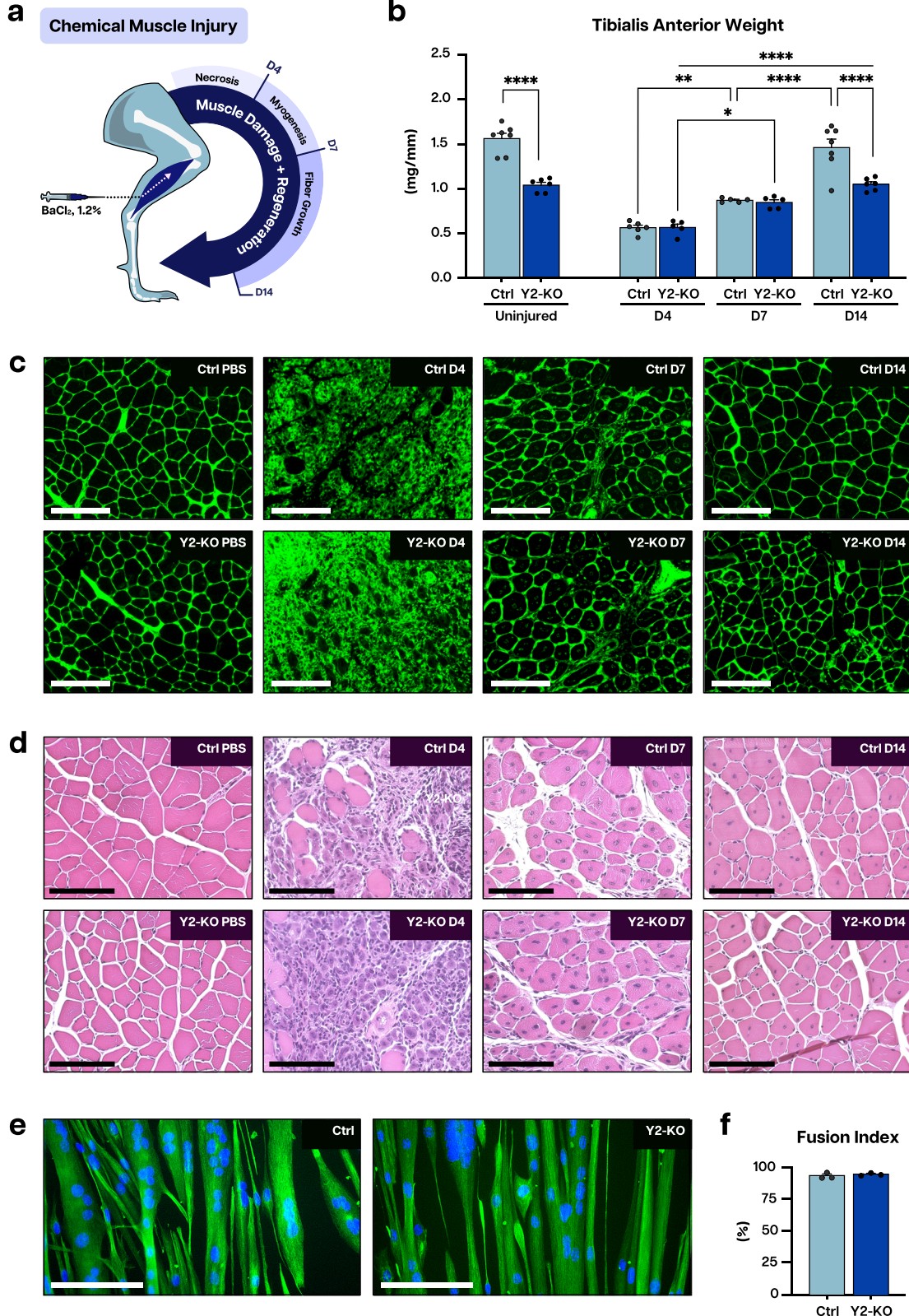

15 m/min, 2 min, 15°; 18, 21, 23, 24 m/min, 1 min, 15°; and +1 m/min, each minute thereafter. Exhaustion was defined as the point at which mice maintained continuous contact with the shock grid for ≥5 s, after which the treadmill and shock grid were ceased.

**Tissue staining and quantification**
Tissues were fixed in 10% neutral buffered formalin solution for 12–24 h, embedded in paraffin, and cut into 5-µm sections prior to

staining with hematoxylin/eosin (H&E) or wheat germ agglutinin (WGA). To assess the myofiber cross-sectional area, prepared sections were deparaffinized, subjected to antigen retrieval for 15 min in boiling sodium citrate buffer (10 mM sodium citrate pH 6.0, 0.05% Tween-20), rinsed in distilled water, incubated for 1 h at room temperature (RT, 22 °C) with blocking buffer (1% BGS in PBS), and incubated for 2 h at RT with wheat germ agglutinin, Alexa Fluor 488 Conjugate (Invitrogen, #W11261, 50 µg/mL). Slides were mounted with VECTASHIELD HardSet

**Fig. 5 | YTHDF2 is dispensable for myogenic regeneration but essential for fiber growth following acute injury. a** Schematic for barium chloride (BaCl$_2$)-induced chemical injury to the tibialis anterior muscle, created in Adobe Illustrator. **b** Weight for Ctrl and Y2-KO tibialis anterior subjected to injection by PBS (uninjured) or BaCl$_2$ 4-, 7-, and 14 days post-injection at 2 months of age. **c** Representative wheat germ agglutinin (WGA, green) staining for Ctrl and Y2-KO tibialis anterior 0-(PBS), 4-, 7-, and 14 days post-injection with BaCl$_2$. **d** Representative hematoxylin/eosin (H&E) staining for Ctrl and Y2-KO tibialis anterior 0- (PBS), 4-, 7-, and 14 days post-injection with BaCl$_2$. **e** Representative myotube (Desmin, green) staining for differentiated myoblasts isolated from Ctrl and Y2-KO total hindlimb, with

counterstain DAPI (blue). **f** Quantification of fusion index for primary myoblast differentiation experiment. Biological animal replicates: $n = 7$ (Ctrl Uninjured), 6 (Y2-KO Uninjured), 6 (Ctrl D4), 5 (Y2-KO D4), 5 (Ctrl D7), 5 (Y2-KO D7), 7 (Ctrl D14), and 6 (Y2-KO D14) for panel **b**. Primary myoblast culture replicates $n = 3$ (Ctrl) and 3 (Y2-KO) for panel **e**, **f**. Data were presented as the mean ± SEM with the individual biological samples shown. Significance was determined by two-way ANOVA with Tukey's HSD multiple-comparison test for comparison of the means of Ctrl and Y2-KO mice across uninjured, and BaCl$_2$-injured D4, D7, and D14 groups: *$p \le 0.05$, **$p \le 0.01$, ****$p \le 0.0001$. Scale bar = 125 μm for panel **c**. Scale bar = 100 μm for panel **d**. Scale bar = 100 μm for panel **e**.

Antifade Mounting Medium (Vector Labs, #H-1400-10) prior to imaging with EVOS Imaging System (Invitrogen, Thermo Fisher Scientific, Waltham, MA, USA). Cross-sectional areas were quantified using ImageJ 1.53k (National Institutes of Health [NIH], Bethesda, MD, USA) as previously described[70].

For fiber-type staining, 10-μm cryosections were prepared from the tibialis anterior. Sections were fixed in 4% paraformaldehyde (PFA) in PBS at RT, permeabilized with 0.3% Triton X. Permeabilized sections were incubated for 1 h at RT with blocking buffer (1% TSA blocking reagent [Perkin Elmer, FP1012], 10% heat-inactivated goat serum in PBS, 0.3% Triton X). Slides were incubated with primary antibodies from the Developmental Studies Hybridoma Bank (DSHB): Myosin heavy chain Type I (DSHB, #BA-D5, 1:40), Myosin heavy chain Type IIA (DSHB, #SC-71, 1:10), Myosin heavy chain Type IIB (DSHB, #BF-F3, 1:40), Myosin heavy chain [all but Type IIX] (DSHB, #BF-35, 1:40), and anti-laminin (Millipore Sigma, #L9393, 1:1000) in blocking buffer overnight at 4 °C. Following secondary antibody incubation for 1 h in blocking buffer, slides were mounted with Fluoromount-G Mounting Medium (Invitrogen, #00-4958-02). The following secondaries were used: Goat anti-Mouse IgG2b Cross-Adsorbed Secondary Antibody, Alexa Fluor 647 (Invitrogen, #A-21242, 1:1000) for DSHB #BA-D5, Goat anti-Mouse IgG1 Cross-Adsorbed Secondary Antibody, Alexa Fluor 647 (Invitrogen, #A-21240, 1:1000) for DSHB #SC-71 and #BF-35, Goat anti-Mouse IgM (Heavy chain) Secondary Antibody, Alexa Fluor 647 (Invitrogen, #A-21238 1:1000) for DSHB #BF-F3, and Goat anti-Rabbit IgG Secondary Antibody, DyLight 488 (Thermo Fisher Scientific, #35552, 1:1000) for laminin. Counterstain DAPI (Invitrogen, #D1306, 0.1 μg/mL) was used for nuclear visualization. Images were obtained using a Zeiss AxioScope microscope and Zeiss AxioCam monochrome charge-coupled device (CCD) camera and AxioVision software SE64 Rel. 4.8 (ZEISS, Dublin, CA, USA). Single-channel, grayscale images were taken, and (pseudo-)color was ascribed, after which channels were exposure matched across all samples and merged in ImageJ 1.53k (National Institutes of Health [NIH], Bethesda, MD, USA).

## Primary myoblast isolation, culture, immunofluorescence staining, and fusion index quantification

Primary myoblasts were isolated from Y2-KO and control animals via magnetic-activated cell sorting (MACS). Hindlimb muscles were minced and enzymatically digested in 750 U/mL Collagenase Type II (Worthington Biochemical, #LS004176) prepared in wash medium (Ham's F10 media, supplemented with 10% horse serum and 1X Pen-Strep) for 90 min in a 37 °C shaking water bath. Digested muscle tissue was washed with wash medium and centrifuged at 500×$g$ for 10 min at 4 °C, then digested with 33 U/mL Collagenase Type II (Worthington Biochemical, #LS004176) and 0.37 U/mL Dispase (Gibco, #17105-041) for 30 min in a 37 °C shaking water bath. Samples were drawn and expelled ten times via 30cc syringe through a 20 G needle, then filtered using a cell 70μm cell strainer, centrifuged at 500×$g$ for 10 min at 4 °C, and resuspended in 2 mL MACS buffer (0.5% bovine serum albumin (Sigma, #A9576-50 mL), 2-mM EDTA prepared in phosphate-buffered saline; sterilized via 0.22-μm syringe filter). Cell suspensions were filtered using a 40-μm cell filter, centrifuged at 500×$g$ for 10 min at 4 °C, and resuspended in 160 μL MACS buffer.

MACS isolation was performed following the manufacturer's instructions and including duplication of the negative cell selection step (Miltenyi Biotec #130-104-268). In brief, 40 μL SC isolation kit beads were added to cell suspensions and incubated for 15 min with shaking on ice. Samples were applied to freshly prepared MACS LS columns (Miltenyi Biotec, #130-042-401), and flowthroughs were collected and centrifuged at 1000×$g$ for 5 min at 4 °C. The negative selection step was repeated once. Cell pellets were then resuspended in 80 μL MACS buffer, to which 20 μL anti-Integrin α-7 microbeads (Miltenyi Biotec, #130-104-261) were added and incubated for 15 min with shaking on ice. Samples were applied to freshly prepared MACS MS columns (Miltenyi Biotec, 130-042-201), and samples were eluted after washes with 1 ml MACS buffer via removal of columns from MACS separator magnet (Miltenyi Biotec, #130-042-102). Eluted myoblasts were centrifuged at 1000×$g$ for 5 min at 4 °C and resuspended in 2 mL myoblast growth medium (Ham's F10 media, supplemented with 20% embryonic stem-cell FBS (Thermo Fisher Scientific, #10439024), 10% horse serum, 1X Pen-Strep, 0.5% chick embryo extract (MP Biomedicals, #092850145), and 2 ng/mL basic fibroblast growth factor (R&D Systems, 233-FB)) and plated on matrigel-coated (Corning, #354248) culture dishes, then cultured at 37 °C in tissue culture incubators with 5% CO$_2$.

For differentiation experiments, myoblasts were counted and seeded in matrigel-coated eight-chamber slides (Thermo Fisher Scientific, #177445) at a density of 45,000 cells per well. Cells were cultured in myoblast growth medium overnight, then switched to differentiation medium (DMEM, supplemented with 2% FBS and 1X Pen-Strep) and cultured for 4 days with daily media changes, prior to fixation with freshly made 2% paraformaldehyde for 10 min at room temperature (RT, 22 °C). Fixed samples were washed with PBS, permeabilized in 0.3% Triton X in PBS for 5 min, and rinsed with PBT (0.05% Triton X in PBS). After permeabilization, samples were incubated in blocking solution (10% heat-inactivated goat serum in PBT) for 60 min, then incubated with anti-Desmin (Thermo Fisher Scientific, #PA5-16705, 1:200) in blocking solution for 2 h at RT. Samples were washed with PBT (3 × 5 min) and incubated with Goat anti-Rabbit IgG Secondary Antibody, DyLight 488 (Thermo Fisher Scientific, #35552, 1:1000) for 60 min at RT. After secondary antibody incubation, samples were counterstained with DAPI (1 μg/mL) for 10 min, then washed with PBT (2 × 5 min), and mounted with Fluoromount-G (Thermo Fisher Scientific, #00-4958-02).

Fluorescent images were obtained using a Zeiss AxioScope microscope and Zeiss AxioCam monochrome charge-coupled device (CCD) camera and AxioVision software SE64 Rel. 4.8 (ZEISS, Dublin, CA, USA). Single-channel, grayscale images were taken, and (pseudo-)color was ascribed, after which channels were exposure matched across all samples and merged in ImageJ 1.53k (National Institutes of Health [NIH], Bethesda, MD, USA). Quantification of Fusion Indices was performed in ImageJ. To quantify the total number of nuclei, monochrome DAPI images were adjusted for brightness and contrast, then thresholded to convert to binary images, and despeckled for noise reduction. The image scale was set according to the microscope's camera's image dimensions, and measurements were set to area. Particle areas were compared to outline and original

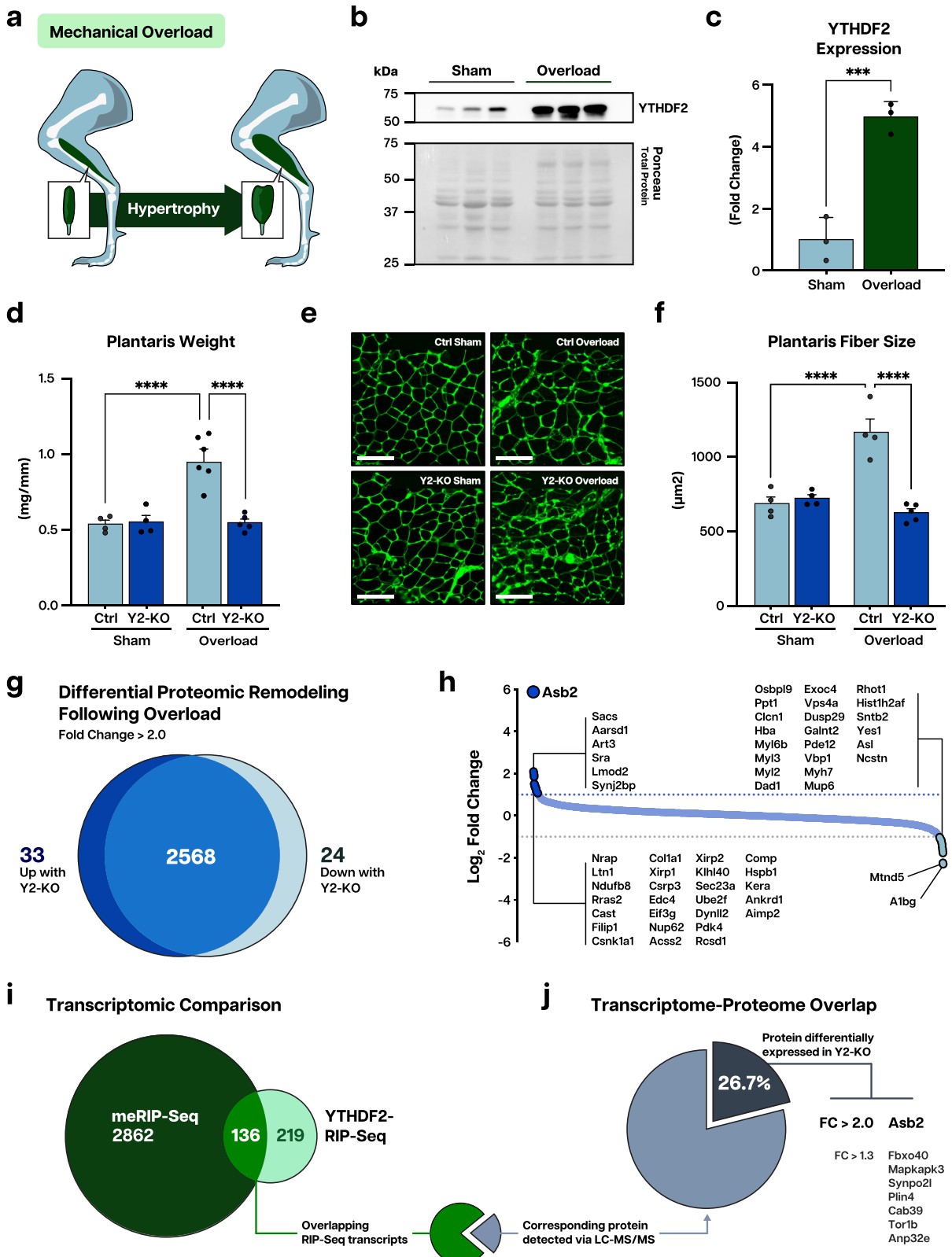

pseudo-colored merged images (Desmin, green; DAPI, blue). To parse individual nuclei from clusters, the average area for all single nuclei was determined and clustered particle areas were divided by this value. To differentiate syncytial and single nuclei, DAPI outline images were merged with original Desmin images. Fusion indices were calculated by dividing the number of fused nuclei by the total number of nuclei per image.

## Muscle measurements and analyses

Plantarflexion torque measurements were performed via 1300 A 3-in-1 Whole Animal System for Mice (Model 1300 A, Aurora Scientific Inc., Canada) as previously described[71]. The right hind paw was taped to the force plate and positioned to align the tibia and foot at 90°. The knee joint was securely clamped at the femoral condyles without compressing the nearby fibular nerve. Two disposable monopolar

**Fig. 6 | YTHDF2 is essential to the hypertrophic response of skeletal muscle.**
**a** Schematic of muscle overload-induced hypertrophy of the plantaris muscle, created in Adobe Illustrator. **b** Western blot analysis of YTHDF2 protein expression in the wild-type plantaris following muscle overload or sham operation.
**c** Quantification of YTHDF2 protein expression using total protein detection by Ponceau stain as a loading control. **d** Plantaris weight for 2-month-old Ctrl and Y2-KO mice 14 days following sham or muscle overload surgery, normalized to tibia length. **e** Representative wheat germ agglutinin (WGA, green) staining and **f** myofiber cross-sectional areas for Ctrl and Y2-KO plantaris 14 days following sham or muscle overload surgery. **g** Venn diagram of LC-MS/MS-identified proteins upregulated in either Ctrl or Y2-KO plantaris muscles (fold change >2.0) following muscle overload. **h** Logarithmic fold change plot for peptides detected in both Ctrl and Y2-KO samples. **i** Venn diagram of transcripts detected with meRIP-Seq and YTHDF2-RIP-Seq. **j** Overlapping transcripts were filtered according to whether the corresponding protein was detected via LC-MS/MS (gray). Proteins with differential expression (fold change >1.3) have been highlighted. Biological animal replicates $n = 3$ (Sham) and 3 (Overload) for panel **b**, **c**; $n = 4$ (Ctrl Sham), 4 (Y2-KO Sham), 6 (Ctrl Overload), and 5 (Y2-KO Overload) for panel **d**; $n = 4$ (Ctrl Sham), 4 (Y2-KO Sham), 4 (Ctrl Overload), and 5 (Y2-KO Overload) for panel **f**; pooled plantaris samples (2 mice per sample) were used for LC/MS, where data reflect $n = 2$ (pooled Ctrl) and 2 (pooled Y2-KO) for panel **g**, **h**. Data are presented as the mean ± SEM with the individual biological samples shown. Significance was determined by two-tailed Student's $t$-test for comparisons between WT sham and overload mice, or by two-way ANOVA with Tukey's HSD multiple-comparison test for comparison of the means of Ctrl and Y2-KO mice following sham or overload surgery: ***$p \leq 0.001$, ****$p \leq 0.0001$. Scale bar = 125 μm for panel **e**.

electrodes (Natus Neurology Inc., Middleton, WI, USA) were subcutaneously inserted over the tibial nerve for stimulation. Maximum plantarflexion tetanic torque (milliNewton-meters [mN·m]) was measured using a series of supramaximal 0.2-ms square-wave stimuli (150 Hz). Muscle fatigue testing was adapted from a previously detailed protocol[72]. After the determination of supramaximal stimulation intensity, mice were subjected to a 5-min fatigue resistance protocol consisting of a total of 150 contractions over a period of 5 min. Each contraction was induced with a 200 ms train of stimuli delivered at 100 Hz with a pulse width of 0.2 ms.

Colorimetric assays were used to detect muscle glycogen (Abcam #ab65620), succinate dehydrogenase (SDH) (abcam #ab228560), and proteasomal activity (Abcam #ab107921) per manufacturer's instruction.

### Western blotting
Protein extracts from whole skeletal muscles were generated using RIPA buffer (150 mM NaCl, 1% nonidet P-40, 0.5% sodium deoxycholate, 0.1% SDS, 25 mM Tris pH 7.4) supplemented with EDTA-free protease (Roche, #11873580001) and phosphatase (Millipore Sigma #524624 and #524625) cocktails. Tissues were snap-frozen in liquid nitrogen and cryopulverized (Cole Parmer Tissue Pulverizer, #40355), sonicated via UCD-500 Bioruptor XL (Denville, NJ, USA) for 10 min (30 s × 320 W, 30 s off). Samples were centrifuged (4 °C × 21,130×g × 20 min) and quantified using Pierce BCA Protein Assay Kit (Thermo Scientific, #23225). Standard Western blotting analysis was performed with the following primary antibodies: YTHDF2 (Abcam, #ab220163, 1:1000), ASB2 (Invitrogen, #PA5-29476, 1:500), SMAD1 (Cell Signaling Technology, D59D7, #6944, 1:1000), SMAD2 (Cell Signaling Technology, D43B4, #5339, 1:1000), SMAD3 (Cell Signaling Technology, C67H9, #9523, 1:1000), SMAD4 (Cell Signaling Technology, D3M6U, #38454, 1:1000), SMAD5 (Cell Signaling Technology, D4G2, #12534, 1:1000), SMAD7 (abcam, #ab216428, 1:1000), SMAD9 (abcam, #ab96698, 1:1000), SMURF2 (Cell Signaling Technology, D8B8, #12024, 1:1000), phosphor-SMAD3 (Abcam, ab52903, 1:1000), FOXO3a (Cell Signaling Technology, D19A7, #12829, 1:1000) phosphor-mTOR (Cell Signaling Technology, D9C2, #5536, 1:1000), mTOR (Cell Signaling Technology, 7C10, #2983, 1:1000), phosphor-AKT (Cell Signaling Technology, D9E, #4060, 1:1000), AKT (Cell Signaling Technology, #9272, 1:1000), LC3I/II (Cell Signaling Technology, D3U4C, #12741 1:1000), Beclin-1 (Cell Signaling Technology, D40C5, #3495, 1:1000), p62 (Cell Signaling Technology, D6M5X, #23214, 1:1000), Ubiquitin (Cell Signaling Technology, E6K4Y, #20326, 1:1000). Membranes were incubated with Peroxidase AffiniPure Goat anti-Rabbit IgG Secondary Antibody (Jackson ImmunoResearch; #111-035-144, 1:10,000) for 90 min at room temperature (22 °C) and imaged via ChemiDoc TOUCH Imaging System (BIO-RAD, Hercules, CA, USA) as previously described[73]. Individual band intensity was quantified using ImageJ 1.53k (National Institutes of Health [NIH], Bethesda, MD, USA), whereby intensity was normalized to the integrated density of total protein loaded, as detected by Ponceau Acid Red 112.

For siRNA-mediated knockdown in H9C2 rat myoblasts (ATCC, # CRL-1446), cells were transfected with control non-targeting siRNA (IDT, #51-01-14-04), si-YTHDF2 TriFECTa DsiRNA Kit (IDT, rn.Ri.Ythdf2.1–3), and/or si-ASB2 TriFECTa DsiRNA Kit (IDT, rn.Ri.Asb2.1–3) with Lipofectamine RNAiMAX (Invitrogen, #13778150). Lysates were collected 48 h, following transfection and prepared as detailed above.

### m⁶A quantification and immunoprecipitation
RNA was extracted from wild-type mouse quadriceps using TRIzol Reagent (Invitrogen, #15596026). $m^6A$ levels were quantified via colorimetric $m^6A$ RNA Methylation Assay Kit (Abcam, #ab185912) per manufacturer's instruction. Extracted RNA (10 μg) was incubated with 5 μg of $m^6A$ antibody (Synaptic Systems, #202003) or control anti-rabbit IgG (Millipore Sigma, #12-370) in ice-cold IPP buffer (150 mM NaCl, 0.1% NP-40, 10 mM Tris-HCl, pH 7.4 and 200 U/mL SUPERase-In RNase Inhibitor (Invitrogen, #AM2696) for 2 h at 4 °C. About 0.5 μg RNA was reserved for input. Pulldown lysates were then incubated with 30 μL Pierce Protein A/G Magnetic Beads (Thermo Scientific, #88803) for 2 h at 4 °C. Samples were washed five times in IPP buffer, and bound RNA was eluted in IPP buffer supplemented with 1 mg/ml Proteinase K (Thermo Scientific, #EO0491) and 0.1% SDS, at 55 °C for 30 min. Isolated RNA was reverse-transcribed using Applied Biosystems High-Capacity cDNA Reverse Transcription Kit (Applied Biosystems, #4368814) as previously described[74]. Selected gene expression differences were analyzed via real-time quantitative polymerase chain (qPCR) using SsoAdvanced SYBR Green Supermix (BIO-RAD, #172527) via CFX Connect (BIO-RAD, Hercules, CA, USA). For comparative transcriptomics, published $m^6A$-immunoprecipitation (meRIP) sequencing data from our group was used[26].

### RNA immunoprecipitation
Wild-type mouse quadriceps were excised, processed via Dounce homogenizer, and sonicated in mild lysis buffer (100 mM Tris pH 7.4, 1 mM EDTA, protease, and RNase inhibitors) via UCD-500 Bioruptor XL (Denville, NJ, USA) for 10 min (30 s × 320 W, 30 s off). Samples were centrifuged (4 °C × 21130 ×g × 30 min), and the supernatants were then incubated with 0.5% NP-40 and 1 mM Dithiothreitol on ice for 15 min. After another round of centrifugation, the supernatants were pre-cleared with Pierce A/G beads coated with 5 μg of anti-rabbit IgG antibody (#12-370, EMD Millipore) for 30 min at 4 °C with rotation and quantified using Pierce BCA Protein Assay Kit (Thermo Scientific, #23225). About 0.3 mg extract was retained for isolation of input RNA. Immunoprecipitation was achieved by incubating 3 mg of protein extract with 5 μg YTHDF2 antibody (Proteintech, #24744-1-AP) or control anti-rabbit IgG (Millipore Sigma, #12-370) at 4 °C overnight. The following day, the samples were incubated with 30 μL of Pierce A/G magnetic beads (#88803, Pierce) for 2 h at 4 °C and washed ten times in wash buffer (100 mM Tris pH 7.4, 50 mM NaCl, 1 mM EDTA,

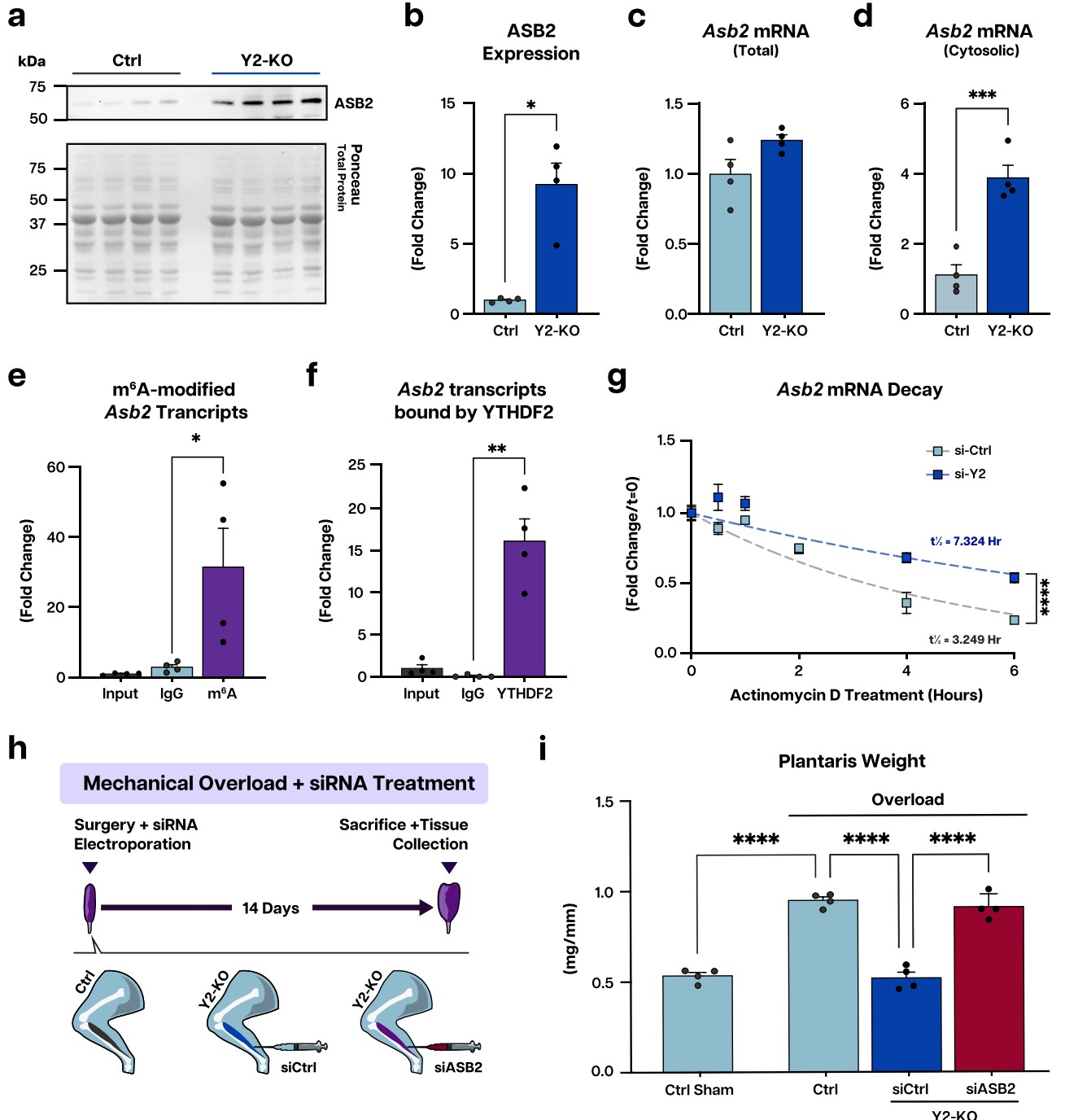

**Fig. 7 | YTHDF2 modulates the expression of E3 ubiquitin ligase ASB2. a** Western blot analysis of ASB2 protein expression in Ctrl and Y2-KO quadriceps at baseline (2 months of age). **b** Quantification of YTHDF2 protein expression using total protein detection by Ponceau stain as a loading control. qPCR analysis of **c** total *Asb2* mRNA and **d** cytosolic *Asb2* mRNA using the quadriceps muscle of Ctrl and Y2-KO mice. qPCR analyses for *Asb2* RNA enrichment in the wild-type quadriceps following RNA immunoprecipitation with **e** m⁶A and **f** YTHDF2, normalized to input RNA. **g** qPCR analysis for *Asb2* following Actinomycin D treatment for the indicated times in H9C2 rat myoblasts transfected with an siRNA pool targeting YTHDF2 (si-Y2) or control non-targeting siRNA (si-Ctrl), normalized to respective $t = 0$ values. Best-fit values for half-life and decay rate were calculated using least squares regression analysis. **h** Schematic of muscle overload-induced hypertrophy of the

plantaris muscle with siRNA intervention, created in Adobe Illustrator. **i** Plantaris weight for Ctrl (sham and overload) and siRNA-treated Y2-KO mice 14 days following muscle overload surgery, normalized to tibia length. Biological animal replicates: $n = 4$ (Ctrl) and 4 (Y2-KO) for panel **a**–**f**, and **i**. Biological cell replicates: $n = 3$ (si-Ctrl at 0, 0.5, 1, and 2 h) or 4 (si-Ctrl at 4 and 6 h), and 4 (si-Y2 at 0, 0.5, 1, 2, 4, and 6 h) for panel **g**. Data were presented as the mean ± SEM with the individual biological samples shown. Significance was determined by a two-tailed Student's $t$-test for comparisons between Ctrl and Y2-KO mice, and between WT IgG and pulldown antibody (m⁶A/YTHDF2). Wilcoxon rank-sum test was used to compare means for nonparametric ASB2 protein quantification data. Grubbs' (ESD) tests were run using GraphPad Prism and outliers were removed from analysis when applicable: *$p \leq 0.05$, **$p \leq 0.01$, ***$p \leq 0.001$, ****$p \leq 0.0001$.

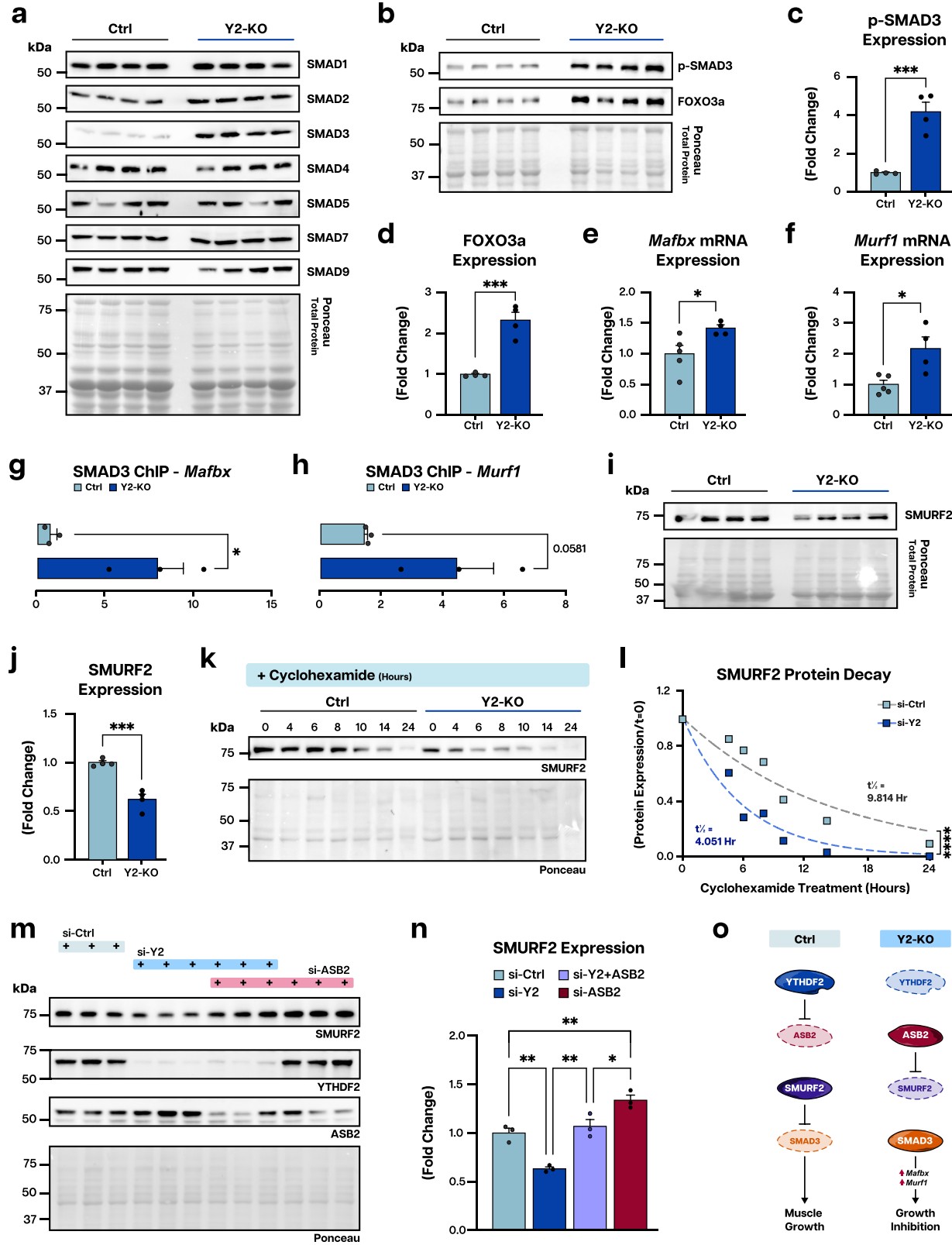

0.1% NP-40). RNA was isolated from the beads via standard phenol-chloroform extraction followed by reverse transcription and qPCR (using CFX Connect [BIO-RAD, Hercules, CA, US]). with gene-specific primers as previously described[74]. For RNA immunoprecipitation and sequencing (RIP-seq), sequencing and analysis of input and immunoprecipitation samples were performed by Novogene (Sacramento, CA, USA) via the Illumina platform.

**RNA stability assay**

mRNA stability was assessed in H9C2 rat myoblasts (ATCC, # CRL-1446) transfected with control non-targeting siRNA (IDT, #51-01-14-04), or si-YTHDF2 TriFECTa DsiRNA Kit (IDT, rn.Ri.Ythdf2.1–3) with Lipofectamine RNAiMAX (Invitrogen, #13778150). Cells were treated with 10 µg/mL transcription inhibitor Actinomycin D (Millipore Sigma, #A1410) 48 h following transfection, and RNA was extracted using

**Fig. 8 | Altered ASB2 expression in the absence of YTHDF2 elicits dysregulated SMAD3 signaling. a** Western blot analysis of protein expression of SMAD proteins 1, 2, 3, 4, 5, 7, and 9 in Ctrl and Y2-KO quadriceps at 2 months of age. **b** Western blot analysis of phosphor-(p-)SMAD3 and FOXO3a protein expression in Ctrl and Y2-KO quadriceps at baseline and **c, d** respective protein expression using total protein detection by Ponceau stain as a loading control. qPCR analysis of **e** *Murf1* and **f** *Mafbx* mRNA expression in Ctrl and Y2-KO gastrocnemius. Chromatin immuno-precipitation (ChIP) analysis of SMAD3 interaction with **g** *MAfbx* and **h** *Murf1* promoters. **i** Western blot analysis of SMURF2 protein expression in Ctrl and Y2-KO quadriceps and **j** quantification of SMURF2 protein expression using total protein as a loading control. **k** Western blot analysis for SMURF2 and **l** quantification following cyclohexamide treatment for the indicated times in H9C2 rat myoblasts transfected with an siRNA pool targeting YTHDF2 (si-Y2) or control non-targeting siRNA (si-Ctrl), normalized to respective $t = 0$ values. Best-fit values for half-life and decay rate were calculated using least squares regression analysis. **m** Western blot analysis for SMURF2, YTHDF2, and ASB2 and **n** quantification following in H9C2 rat myoblasts transfected with an siRNA pool targeting YTHDF2 (si-Y2) and/or ASB2 (si-ASB2), or control non-targeting siRNA (si-Ctrl). **o** Schematic of YTHDF2-mediated muscle growth modulation, created in Adobe Illustrator. Biological animal replicates: $n = 4$ (Ctrl) and 4 (Y2-KO) for panels **a**–**c, d, I, j**; $n = 5$ (Ctrl) and 4 (Y2-KO) for panel **e, f**; $n = 3$ (Ctrl) and 3 (Y2-KO) for panel **g, h, m, n**. Pooled cell replicates were used for panels **k** and **l**. Data were presented as the mean ± SEM with the individual biological samples shown. Significance was determined by a two-tailed Student's *t*-test for comparisons between Ctrl and Y2-KO mice (using respective control values for panel b data): \*$p \leq 0.05$, \*\*$p \leq 0.01$, \*\*\*$p \leq 0.001$, \*\*\*\*$p \leq 0.0001$.

TRIzol Reagent (Invitrogen, #15596026) after 0, 0.5, 1, 2, 4, and 6 h Actinomycin D exposure, followed by reverse transcription and qPCR analysis of Asb2 transcript levels using the following primers: rat *Asb2* 5′-GCACTTCAGCGCTCTACTTC-3′ and 5′-ATGTAGGCGTCGATGTTTG C-3′; via CFX Connect (BIO-RAD, Hercules, CA, USA). The best-fit values for mRNA decay rate ($k$) and half-life were calculated via non-linear least squares regression curve fitting (1 phase decay).

### mRNA analysis by real-time PCR
RNA was extracted using TRIzol Reagent (Invitrogen, #15596026) and reverse-transcribed by Applied Biosystems High-Capacity cDNA Reverse Transcription Kit (Applied Biosystems, #4368814) as previously described[74]. Selected gene expression differences were analyzed via real-time quantitative polymerase chain (qPCR) using SsoAdvanced SYBR Green Supermix (BIO-RAD, #172527) via CFX Connect (BIO-RAD, Hercules, CA, USA). Quantified mRNA expression was normalized to housekeeping gene Rpl7 (ribosomal protein L7) (or ribosomal 5S RNA for RNA immunoprecipitation studies), and expression was presented relative to control levels using the ΔΔCT method of analysis. The following primers were used: mouse *Ythdf2* 5′-TGGTTCTGTGCATCAAAAGGA-3′ and 5′-CACCTCCAGTAGACCAAGC A-3′; mouse *Murf1* 5′-GCTGGTGGAAAACATCATTGACAT-3′ and 5′-CAT CGGGTGGCTGCCTTT-3′; mouse *Mafbx* 5′-CTTTCAACAGACTGGA CTTCTCGA-3′ and 5′-CAGCTCCAACAGCCTTACTACGT-3′; mouse *Rpl7* 5′-TGGAACCATGGAGGCTGT-3′ and 5′-CACAGCGGGAACCTTTTTC-3′; mouse *5 S* 5′-CTACGGCCATACCACCCTG-3′ and 5′-CCTACAGCACCCG GTATTCC-3′; mouse *Asb2* 5′-GCAGAGAACACCTGGATTGCCT-3′ and 5′-TTGGCGTCTGCGTTGTATCGCA-3′; rat *Asb2* 5′-GCACTTCAGCGCTC-TACTTC-3′ and 5′-ATGTAGGCGTCGATGTTTGC-3′; mouse *Smad3* 5′-GCTTTGAGGCTGTCTACCAGCT-3′ and 5′-GTGAGGACCTTGACAAGCC ACT-3′.

### Subcellular RNA fractionation
Muscle lysates were prepared via Dounce homogenizer in lysis buffer (50 mM Tris-HCl, pH 7.6, 50 mM NaCl, 5 mM MgCl$_2$, 0.1% IGE-PAL, 1 mM β-mercaptoethanol, protease, and RNase inhibitors) and were then incubated on ice for 30 min followed by centrifugation (4 °C × 2000×$g$ × 3 min). The supernatant, containing the cytosolic compartment, was retained, and RNA was extracted using standard phenol-chloroform isolation. RNA was reverse-transcribed by Applied Biosystems High-Capacity cDNA Reverse Transcription Kit (Applied Biosystems, #4368814) as previously described[74].

### Chromatin immunoprecipitation (ChIP)
Quadriceps were excised, minced, and crosslinked in 1.5% for-maldehyde in PBS for 15 min at room temperature (RT, 22 °C) with rotation. Crosslinking was quenched by the addition of glycine to a final concentration of 0.125 M, and rotation was continued for 5 min. Samples were centrifuged (4 °C × 1000×$g$ × 5 min), and crosslinked tissues were washed twice with PBS. Lysates were prepared via Dounce homogenizer in ChIP lysis buffer (50 mM HEPES-KOH pH 7.5, 140 mM

NaCl, 1 mM EDTA pH 8, 1% Triton X-100, 0.1% sodium deoxycholate, 0.1% SDS, protease inhibitors), treated with micrococcal nuclease (Roche, #10107921001) at 37 °C for 15 min (+CaCl$_2$, deactivated by EGTA) sonicated via Fisherbrand Model 50 Sonic Dismembrator (Thermo Fisher Scientific, Waltham, MA, USA) for 8 min (30 s at 60% amplitude, 30 s off), and cleared via centrifugation (4 °C × 8000×$g$ × 10 min). Supernatants were diluted in RIPA buffer (150 mM NaCl, 1% nonidet P-40, 0.5% sodium deoxycholate, 0.1% SDS, 25 mM Tris pH 7.4) supplemented with protease inhibitors (Roche, #11873580001). Samples were then incubated with SMAD3 antibody (Cell Signaling Technology, C67H9, #9523, 1:1000) or control anti-rabbit IgG (Millipore Sigma, #12-370) (10 μg per 25 μg DNA) for 1 h at 4 °C with rotation. Pierce Protein A/G Magnetic Beads (Thermo Scientific, #88803) were added to each sample and rotated at 4 °C overnight. Immunoprecipitated samples were washed once each with high-salt (0.1% SDS, 1% Triton X-100, 2 mM EDTA, 20 mM Tris-HCl pH 8.0, 500 mM NaCl), low-salt (0.1% SDS, 1% Triton X-100, 2 mM EDTA, 20 mM Tris-HCl pH 8.0, 150 mM NaCl), and LiCl wash buffer (0.25 M LiCl, 1% NP-40, 1% sodium deoxycholate, 1 mM EDTA, 10 mM Tris-HCl pH 8.0). Elution buffer (1% SDS, 100 mM NaHCO$_3$) was added to the beads and incubated for 15 min at 30 °C. RNAse A (+NaCl) and samples were incubated at 65 °C for 4 h. Proteinase K (Thermo Scientific, #EO0491) was added to the eluted DNA, and samples were incubated at 55 °C for 1 h. DNA was purified via standard phenol-chloroform extraction followed by reverse transcription and qPCR with gene-specific primers as previously described[74].

Gene expression differences were analyzed via real-time quantitative polymerase chain (qPCR) using SsoAdvanced SYBR Green Supermix (BIO-RAD, #172527) via CFX Connect (BIO-RAD, Hercules, CA, USA). via CFX Connect (BIO-RAD, Hercules, CA, USA). mRNA expression was normalized to respective IgG and expression was presented relative to control levels using the ΔΔCT method of analysis. Primers were designed to flank a published FOXO3a binding element upstream of the *Murf1* coding sequence[75] and SMAD3 binding element consensus motif CCAGACA we identified upstream of *Mafbx*[76]. The following primers were used: mouse *Murf1* 5′-CTGGGCCTCTGCAC CTG-3′ and 5′-TGGACACACTTGTCACCTGG-3; mouse *Mafbx* 5′-AGCC TCAGTTCTTGGGCTCT-3′ and 5′- CGGGCTTCCTTGAGTGTCTT-3′.

### Protein stability assay
Protein stability was assessed in H9C2 rat myoblasts (ATCC, # CRL-1446) transfected with control non-targeting siRNA (IDT, #51-01-14-04), si-YTHDF2 TriFECTa DsiRNA Kit (IDT, rn.Ri.Ythdf2.1−3), or si-ASB2 TriFECTa DsiRNA Kit (IDT, rn.Ri.Asb2.1−3) with Lipofectamine RNAi-MAX (Invitrogen, #13778150). Cells were treated with 50 μg/mL protein synthesis inhibitor cyclohexamide (Millipore Sigma, #C4859) 48 h following transfection. Protein extracts were generated using RIPA buffer (150 mM NaCl, 1% nonidet P-40, 0.5% sodium deoxycholate, 0.1% SDS, 25 mM Tris pH 7.4) supplemented with protease and phosphatase inhibitors (Roche, #11873580001; Millipore Sigma, #524624 and #524625) after 0, 4, 6, 8, 10, 14, and 24 h of cycloheximide

exposure. Following sonication via UCD-500 Bioruptor XL (Denville, NJ, USA) for 10 min (30 s × 320 W, 30 s off), centrifugation (4 °C × 21,130×$g$ × 20 min), and protein quantification, standard Western blotting analysis was performed with anti-SMURF2 (Cell Signaling Technology, D8B8, #12024, 1:1000). Membranes were incubated with Peroxidase AffiniPure Goat anti-Rabbit IgG Secondary Antibody (Jackson ImmunoResearch; #111-035-144, 1:10,000) for 90 min at room temperature (22 °C) and imaged via ChemiDoc TOUCH Imaging System (BIO-RAD, Hercules, CA, USA) as previously described[73]. Individual band intensity was quantified using ImageJ 1.53k (National Institutes of Health [NIH], Bethesda, MD, USA), whereby intensity was normalized to the integrated density of total protein loaded, as detected by Ponceau S (Acid Red 112). The best-fit values for protein decay rate ($k$) and half-life were calculated via non-linear least squares regression curve fitting (1 phase decay).

## Mass spectrometry-based proteomics analysis

Following overload, pooled plantaris samples from control or Y2-KO ($n$ = 2 mice per lysate) were prepared in LC-MS/MS lysis buffer (50 mM Tris-HCl, pH 7.6, 150 mM NaCl, 0.5% sodium deoxycholate, 1% SDS, 1 mM EDTA). Samples were sonicated via UCD-500 Bioruptor XL (Denville, NJ, USA) for 10 min (30 s × 320 W, 30 s off) and cleared (4 °C × 21,130×$g$ × 20 min), and protein was quantified via Pierce BCA Protein Assay Kit (Thermo Scientific, #23225). Protein quality was assessed via silver stain, and 100 μg extracted protein was submitted to The Ohio State University CCIC Mass Spectrometry and Proteomics (MSP) Facility. Tandem-mass tag analysis was performed by labeling all samples with reactive isobaric tags, mixed and analyzed in a single liquid chromatography-mass spectrometry (LC-MS/MS) experiment as previously described[77].

## Statistical analysis

All results are presented as mean ± SEM, with dots indicating individual replicates within a group. Data normality was assessed via the Shapiro–Wilk test, after which statistical analysis between two groups was performed via unpaired two-tailed $t$-test for normally distributed groups. Wilcoxon rank-sum test was used to compare group means for nonparametric value sets. Comparisons across two genotypes and two experimental conditions were analyzed via two-way ANOVA followed by Tukey's HSD multiple-comparison test. Comparisons across three groups were analyzed via one-way ANOVA. Grubbs' (ESD) tests were run using GraphPad Prism, and outliers were removed from analysis when applicable. $p$ value of ≤0.05 (*) was considered significant for all described tests, where **$p$ ≤ 0.01, ***$p$ ≤ 0.001, ****$p$ ≤ 0.0001; $p$ values are included in Supplementary Data 3. Analyses were conducted using GraphPad Prism Ver 9.4.0 (GraphPad Software). ImageJ 1.53k (National Institutes of Health [NIH], Bethesda, MD, USA) and Image Lab Ver 6.10 (BIO-RAD, Hercules, CA, USA) were used for Western blot quantification. CFX Maestro (BIO-RAD, Hercules, CA, USA) and Microsoft Excel Ver 16.72 (Microsoft, Redmond, WA, USA) were used for qPCR analysis.

## Reporting summary

Further information on research design is available in the Nature Portfolio Reporting Summary linked to this article.

## Data availability

Proteomics data are available through MassIVE (https://massive.ucsd.edu/ProteoSAFe/dataset.jsp?task=fec01c55fda44fb999d0e9e58845e6a6). m⁶A sequencing analyses are available through Gene Expression Omnibus (https://www.ncbi.nlm.nih.gov/geo/query/acc.cgi?acc=GSE179368). Source data are provided with this paper.

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

## Acknowledgements

This work was supported by the NIH under grants [R01 HL 136951 and R01 HL 154001] to F.A., [F31 HL 158234-01] to C.J.G, and R01 AR078231 to C.L. We express our gratitude to the CCIC Mass Spectrometry and Proteomics (MSP) Facility at the Ohio State University with NIH support grants S10 OD025008 and P30CA016058.

## Author contributions

C.J.G. and F.A. conceived the project. C.J.G. and F.A. wrote and revised the manuscript. C.J.G. performed most of the in vivo and ex vivo studies, including animal surgeries, animal injections, animal exercise testing, tissue harvesting and collection, protein and RNA extractions, Western blotting, real-time PCR, and histological analyses. C.P.R. performed transcriptomic and proteomic analyses and m6A immunoprecipitations. V.A.G. performed RNA immunoprecipitations and RNA and protein stability assays. M.W., A.K., and W.D.A. performed muscle force and fatigue testing. K.S. and C.L. performed myofiber isolation, and K.S. performed fusion index assessment. C.L. supervised fiber typing analyses.

## Competing interests

The authors declare no competing interests.
