## [Peer Review File · Nature Communications]

YTHDF2 governs muscle size through a targeted modulation of proteostasisREVIEWER COMMENTS

Reviewer #1 (Remarks to the Author):

In this manuscript the authors investigate the role of the m6A reader protein Ythdf2 in skeletal muscle tissue. The authors found that skeletal muscle specific deletion of Ythdf2 is associated with impaired muscle growth and finally link Ythdf2 to the expression of the ubiquitin ligase ASB2. While the manuscript is of general interests, the experiments and data presentation sound, the link between Ythdf2 and Asb2 could be characterized in more detail.

Major concern:

1) Very recently the role of Ythdf2 in cardiac muscle tissue has been described by two groups. Here, lack of Ythdf2 rather drives hypertrophic growth in KO mouse models compared to WT mice. What drives the molecular difference between impaired growth and hypertrophic growth in skeletal and cardiac muscle tissue? A detailed comparative analysis of identified target would help to clarify those differences.

2) What drives the massive increase in expression of Ythdf2 in response to hypertrophic growth? Is this regulated by a specific transcription factor? Is Ythdf2 sufficient to increase muscle size both in vitro and in vivo?

3) The authors should formally show that Smurf is regulated by Asb2 via protein stability.

4) How are overall m6A levels in the Ythdf2 KO mice? How many m6A containing transcripts are changed in expression in the KO mice and how is the overlap to the proteomic data?

Reviewer #2 (Remarks to the Author):

The present study analysed the role of YTHDF2 in muscle mass regulation. The authors generated a muscle specific knockout and showed reduced muscle mass maintenance and growth during adulthood and in regenerating or overloaded muscles, respectively. They further showed a link between YTHDF2 and expression of an ubiquitin ligase, ASB2B, which has been linked to muscle wasting. Inhibition of ASB2B rescued the defect of muscle growth during plantaris overload. The identification of YTHDF2 as a regulator of muscle mass and the connection with ASB2B is of interest and novel. The major limitations

are the mechanistic insights. It is unlikely that just expression of ASB2B triggers a late atrophy response (8 months of age) but an early myofiber growth defect in regenerating muscles. The latter point is also in contradiction with the data of muscles mass at birth which show no difference and no growth defect during the first 30 days after delivery. The mechanisms of muscle hypertrophy or age-related muscle wasting and muscle regeneration may be completely different. Also, the insight that regulate ASB2B expression a must be better characterized. The authors should consider the following points

1) Authors should also test fatigue during the force measurement experiments. This piece of data would better explain the reduced physical activity.

2) Show also fiber type distribution in terms of SDH or COX staining. This because the decrease in physical performance during treadmill can not be explained just with a reduced muscle mass or muscle force but with a shift in a more glycolytic metabolism. In case that there is no shift of SDH/COX staining, then mitochondrial function and glycogen content must be analysed to explain exercise intolerance.

3) The changes of muscle mass at 8 months of age should be better characterized. Authors should check whether protein synthesis, protein ubiquitination or autophagy system are dysregulated by the absence of YTHDF2.

4) Show ASB2B expression in muscles of YTHDF2 KO at 8 months and in regenerating muscles at 7 days. A transcriptomic and proteomic analyses of muscles at 8 months of age would be useful to understand the phenotype and to identify how much similar are the profiles when compared to overloaded muscles.

5) Show that ASB2B inhibition rescue the defect in muscle mass in YTHDF2 KO at 8 months and in regenerating muscles.

6) YTHDF2 should promote the mRNA decay, while data suggest the enhance translation (the increase of ASB2B mRNA is modest and not significant Fig 6C). However, authors connected ASB2B protein increase with an increased stability of the transcript. This finding is supported by in vitro data with actinomycin D. However, this hypothesis does not match with the in vivo data (no increase of transcript and 10 fold increase of protein). Authors should check protein translation by immunoprecipitating ribosomes and checking the abundance of YTHDF2 transcript and by performing in vitro translational assay.

7) Is the defect in muscle recovery during regeneration due to an abnormal myoblast fusion and nuclear accretion or is due to a defect in protein synthesis during myotube growth?

8) The link with ASB2B is of interest but the connection with SMAD2/3 is weak. Just changing the level of expression of Smad3 may be not sufficient to explain the effect on muscle mass. Please show the level of p-Smad2/3 and p-Smad1/5/8 and a ChIP experiment is mandatory to identify whether MuRf1 and Atrogin1 expression is under Smad control in YTHDF2 KO.

9) Because Smad2/3 has been linked to AKT-FoxO pathway also this signaling must be checked.

10) Finally, rescue experiment in which YTHDF2 is re-expressed in the knockout mice in adulthood and check how much of muscle wasting is reverted is mandatory.

Reviewer #3 (Remarks to the Author):

This work reveals the function of m6A reader YTHDF2 in skeletal muscles regulating muscle size. Mechanistically, the author demonstrates that YTHDF2 binds and promotes the decay of m6A modified ASB2 mRNA, which inhibits muscle growth. Knock out of YTHDF2 results in increased ASB2 protein level to further activate the growth inhibition pathway, leading to muscle atrophy in postnatal mice and failure of muscle hypertrophy after overload. The study offers some novel insights on how YTHDF2 functions in muscle growth. I have the following concerns.

Major comments:

1. As a m6A reader, YTHDF2 should have many targets other than ASB2. To fully understand the function of YTHDF2, it is necessary to conduct MeRIP-seq and YTHDF2 RIP-seq or CLIP-seq.

2. In figure 6 and figure 7, why is H9C2 rat myoblast used for the in vitro experiments? This is derived from embryonic BD1X rat heart tissue. The author reports that postnatal heart muscle weight is not affected after YTHDF2 knock out, suggesting YTHDF2 may not regulate ASB2 mRNA decay in heart muscle. Since the animal experiments were performed in mice limb muscles, it is more reasonable to use freshly isolated primary myoblast or at least C2C12 mouse myoblast cells for these experiments.

Minor comments:

1. In page 4, “and histopathological analysis showed no overall disruption of muscle architecture (Fig. k).”, there is no figure K in figure 1.

Reviewer #1

In this manuscript the authors investigate the role of the m6A reader protein Ythdf2 in skeletal muscle tissue. The authors found that skeletal muscle specific deletion of Ythdf2 is associated with impaired muscle growth and finally link Ythdf2 to the expression of the ubiquitin ligase ASB2. While the manuscript is of general interests, the experiments and data presentation sound, the link between Ythdf2 and Asb2 could be characterized in more detail.

We thank this reviewer for their work on our manuscript. We believe we now further strengthened the link between YTHDF2 and ASB2 by following the reviewers' suggestions. Indeed, we added unbiased approaches through sequencing of transcripts pulled down with YTHDF2 in muscle and cross-analyses with m⁶A immunoprecipitation sequences and proteomics data. The results reinforced the importance of ASB2 as target of YTHDF2 (please see specific answer to comment #4 for more details).

Major concern:

1) Very recently the role of Ythdf2 in cardiac muscle tissue has been described by two groups. Here, lack of Ythdf2 rather drives hypertrophic growth in KO mouse models compared to WT mice. What drives the molecular difference between impaired growth and hypertrophic growth in skeletal and cardiac muscle tissue? A detailed comparative analysis of identified target would help to clarify those differences.

This is an interesting question. To address this, we compared the differentially expressed proteins in our cardiac and skeletal muscle YTHDF2-KO versus controls datasets and found only 3 overlapping proteins (figure below for reviewer only), suggesting disparate, context-dependent mechanisms by which YTHDF2 regulates these tissue types. Of note, of the 3 overlapping proteins (MYL6B, MYH7, and MtND5), MYL6B and MYH7 show opposite directionality of change (increasing in cardiac and decreasing in skeletal muscle samples), while MtND5 is not derived from nuclear encoded transcripts and therefore unlikely to be directly regulated by YTHDF2. The fact that the YTHDF2-dependent proteomic remodeling differs significantly between the heart and skeletal muscle could at least in part explain the resulting different phenotypes that deficiency in YTHDF2 triggers across tissues.

Reviewer Only Figure. Venn diagram of differentially expressed gene products between the heart and skeletal muscle in the absence of YTHDF2. Proteins were identified via LC-MS/MS (fold change>2.0).

2) What drives the massive increase in expression of Ythdf2 in response to hypertrophic growth? Is this regulated by a specific transcription factor? Is Ythdf2 sufficient to increase muscle size both in vitro and in vivo?

Thanks for these important questions. We have now found that reintroduction of YTHDF2 via adeno-associated viruses (AAV) to Y2-KO muscles restores baseline muscle mass (**new Fig. 1l-m**), though AAV-YTHDF2 does not alone increase muscle size of wildtype unstressed mice (**new Supplementary Fig. 1**). These data suggest that, though indispensable for muscle growth, in the tested conditions YTHDF2 is not sufficient to stimulate hypertrophy. Promoter sequence analysis for Ythdf2 (using Genehancer regulatory Elements) detected binding domains for Teads1-4 and JunB, leading to interesting speculations on an upstream regulatory role for these factors. However, the increase in YTHDF2 expression following hypertrophic stimuli is likely a component of a pro-growth molecular remodeling network, and other players may be needed to act in concert with YTHDF2 to increase muscle size above wildtype levels.

3) The authors should formally show that Smurf is regulated by Asb2 via protein stability.

The reviewer raises another good point. We have now performed a protein stability assay and observed enhanced stability of SMURF2 in the absence of ASB2 (**new Supplementary Fig. 7**). Thank you for helping us to strengthen our proposed mechanism.

4) How are overall m6A levels in the Ythdf2 KO mice? How many m6A containing transcripts are changed in expression in the KO mice and how is the overlap to the proteomic data?

We thank the reviewer for this comment. We have now quantified m⁶A content relative to total adenosine (m⁶A/A) by ELISA in control and Y2-KO muscle (**new Fig. 1k**). We observed no change in global m⁶A content, allowing for us to attribute our phenotype to YTHDF2's regulation of m⁶A, rather than altered modification levels. To better understand the relation between transcriptomic and proteomic alterations, we performed YTHDF2-RIP-Seq and compared the detected transcripts with those found with m⁶A (me-)RIP-Seq (**new Fig. 6i**). Of the overlapping transcripts, we assessed whether the corresponding protein was detected by our LC-MS/MS analysis and what proportion was ultimately affected by YTHDF2 at the protein level. We found that 26.7% of these proteins were differentially expressed in the absence of YTHDF2, with ASB2 confirmed as the most robust target (**new Fig. 6j**).

We believe the new data presented in the manuscript in response to this reviewer's comments significantly reinforce our study and we are therefore grateful for their help.

Reviewer #2 (Remarks to the Author):

The present study analysed the role of YTHDF2 in muscle mass regulation. The authors generated a muscle specific knockout and showed reduced muscle mass maintenance and growth during adulthood and in regenerating or overloaded muscles, respectively. They further showed a link between YTHDF2 and expression of an ubiquitin ligase, ASB2B, which has been linked to muscle wasting. Inhibition of ASB2B rescued the defect of muscle growth during plantaris overload. The identification of YTHDF2 as a regulator of muscle mass and the connection with ASB2b is of interest and novel. The major limitations are the mechanistic insights. It is unlikely that just expression of ASB2B triggers a late atrophy response (8 months of age) but an early myofiber growth defect in regenerating muscles. The latter point is also in contradiction with the data of muscles mass at birth which show no difference and no growth defect during the first 30 days after delivery. The mechanisms of muscle hypertrophy or age-related muscle wasting and muscle regeneration may be completely different. Also, the insight that regulate ASB2B expression a must be better characterized.

We appreciate this reviewer's comments and help with our revision. In response to the raised concerns, we present a significantly revised study and we believe the new data further clarifies the mechanism behind the observed phenotypes. As specified in more details below in response to specific points, we now provide an overview of catabolic and anabolic pathways at play in the various tested conditions, which showed increased catabolic markers as common denominators. We also tested the fusion index of myoblast isolated from control and YTHDF2 deficient muscles. This analysis showed no overt defect in this process, which plays a dominant role in the first 4 weeks of postnatal murine muscle growth and in the first 7 days post BaCl₂ injury in adult mice, conditions where we see no differences between genotypes. Further, we show that ASB2 expression is elevated with YTHDF2 deficiency in all tested conditions; we also present additional insights on the regulation of ASB2 and we show increases in its mRNA abundance in the cytoplasm of YTHDF2-KO muscle using subcellular fractionations, supporting the role of YTHDF2 in controlling the stability of Asb2 transcripts after their nuclear export. We agree that the suggested analyses were much needed, and we are grateful to this reviewer for helping us improve the study.

The authors should consider the following points

1) Authors should also test fatigue during the force measurement experiments. This piece of data would better explain the reduced physical activity.

We have now completed a muscle fatigue protocol and incorporated the fatigue curve (in output torque) as **new Supplementary Fig. 3a**. While the curve shows an initial defect in maximal output, consistent with our tetanic stimulation data (**Fig. 3n**), we observed no significant change in the fatiguability of Y2-KO mice following prolonged stimulation. This was an interesting observation for us, and we continued assessment in line with your other suggestions (detailed below) to attempt to better understand the observed performance defect.

2) Show also fiber type distribution in terms of SDH or COX staining. This because the decrease in physical performance during treadmill can not be explained just with a reduced muscle mass or muscle force but with a shift in a more glycolytic metabolism. In case that there is no shift of SDH/COX staining, then mitochondrial function and glycogen content must be analysed to explain exercise intolerance.

We have now measured SDH activity and glycogen content, where we observe trends toward a reduction in Y2-KO muscles (**new Supplementary Fig. 3b-c**). Though not significant, it is important to note that these findings are normalized to total load and the difference in mass for each genotype may mitigate differences in total abundance/activity if considering that the available total muscle mass is smaller for Y2-KO mice. We also recognize the potential for contribution of other metabolic networks. Indeed, a previous study on ASB2, our key

downstream target for YTHDF2, identified perturbation of regulators of beta oxidation as consequence of the catabolic cascade initiated by this factor [PMID: 33516941]. We now characterized proteostasis in our model more globally as described in our answer to the following comment and have addressed the potential link to metabolism in the discussion section as follows: “... while we detected no significant shift in glycogen concentration or SDH activity in Y2-KO muscles, our findings do not preclude the contribution of other metabolic cascades to our phenotype. Indeed, previous reports have identified a role for ASB2 in fatty acid β -oxidation ...”

3) The changes of muscle mass at 8 months of age should be better characterized. Authors should check whether protein synthesis, protein ubiquitination or autophagy system are dysregulated by the absence of YTHDF2.

We have now incorporated these assessments in a new figure, where we present the molecular signature of 8-month-old muscles (**new Fig. 4**). We detected no significant disruption in protein synthesis pathways, such as AKT or mTOR (**new Fig. 4a-g**). Conversely, we observed significant upregulation of autophagic markers LC3II and Beclin-1, as well as increased polyubiquitination in Y2-KO muscles (**new Fig. 4i-j** and **4l-m**). We further tested for proteasomal activity and found this to be enhanced in Y2-KO muscles (**new Fig. 4o**). Overall, these data illustrate a shift in favor of catabolic signaling in the absence of YTHDF2. We thank the reviewer for this suggestion that has greatly added to our understanding of the observed phenotype.

4) Show ASB2B expression in muscles of YTHDF2 KO at 8 months and in regenerating muscles at 7 days. A transcriptomic and proteomic analyses of muscles at 8 months of age would be useful to understand the phenotype and to identify how much similar are the profiles when compared to overloaded muscles.

The suggested assessments have now been included as **new Supplementary Fig. 5**. We report a significant increase in ASB2 expression both in 8-month-old muscle (**new Supplementary Fig. 5a-b**) and muscles 7 days following acute injury (**new Supplementary Fig. 5c-d**). Together, our data suggest an upregulation of ASB2 in Y2-KO muscle from all tested conditions. We also now present a molecular analysis of proteostasis pathways both at 8 months of age (**new Fig. 4**) and at 7 days following BaCl₂ injury (**new Supplementary Fig. 4**), showing protein degradation markers as consistently increased with YTHDF2 deficiency (results explained in more details in answer #7).

5) Show that ASB2B inhibition rescue the defect in muscle mass in YTHDF2 KO at 8 months and in regenerating muscles.

Given the route of administration used for our ASB2 inhibition (i.e., intramuscular electroporation of siRNA), we were unable to complete the suggested studies. Given short half-life of siRNAs, the effects of inhibition would be limited to ~2 weeks, thus requiring multiple administrations for 8-month-old mice. This process requires permeabilization of the muscle and electroporation of the injected siRNA with each administration, which could result in tissue damage, necrosis, or rhabdomyolysis with this constant insult [PMC122059, PMID11708877]. Relatedly, inhibition of ASB2 in regenerating muscle would require permeabilization/ electroporation in addition to injection of barium chloride; electroporation is best effective on small muscles such as EDL, FDB and plantaris, while our regeneration data is on tibialis anterior, further complicating experimental design and data reproducibility. As detailed above, however, we have now shown aberrant ASB2 expression in Y2-KO mice across all experimental conditions (**new Supplementary Fig. 5**), suggesting a pivotal role for YTHDF2 in its regulation irrespective of context.

6) YTHDF2 should promote the mRNA decay, while data suggest the enhance translation (the increase of ASB2B mRNA is modest and not significant Fig 6C). However, authors connected ASB2B protein increase with an increased stability of the transcript. This finding is supported by in vitro data with actinomycin D.

However, this hypothesis does not match with the in vivo data (no increase of transcript and 10 fold increase of protein). Authors should check protein translation by immunoprecipitating ribosomes and checking the abundance of YTHDF2 transcript and by performing in vitro translational assay.

This is a very good point. Published work suggests YTHDF2 stimulates mRNA decay in the cytosol [PMID27558897, PMID24284625]. Since translation is also an exclusively cytoplasmic process, we reasoned that subcellular fractionation of our samples could help clarify the impact of YTHDF2 on mRNA abundance. Indeed, as site of transcription, the nucleus contains high levels of mRNA, which could confound the results when using total RNA extracts. To reconcile the disconnect in total mRNA and protein abundance, we performed subcellular fractionation of skeletal muscle lysates followed by qPCR for *Asb2* transcripts. We observed a significant increase in *Asb2* mRNA in the cytosolic compartment of Y2-KO skeletal muscle compared to control littermates (**new Fig. 7d**). These data are in agreement with a post-transcriptional role of YTHDF2 in the cytoplasm and reinforces the in vitro mRNA decay result presented in **Fig. 7g**.

7) Is the defect in muscle recovery during regeneration due to an abnormal myoblast fusion and nuclear accretion or is due to a defect in protein synthesis during myotube growth?

The reviewer raises another excellent point. We have since performed myoblast isolation and fusion index assessment and observed no defect in myotube formation in the absence of YTHDF2 (**new Fig. 5e-f**). Since myoblast fusion is a key growth mechanism for murine muscle during the first month of life [PMID23612709] as well as during the first week following chemical injury [PMID23612709, PMID32456017], we believe this new data helps explaining the lack of phenotype at 1 month of age (**Fig. 1e-g**) as well as the ability of Y2-KO muscle to respond to BaCl₂ injury equally well as their controls at the day 7 time point (**Fig. 5b-d**). Further, to better understand the balance of synthesis and decay in regeneration, we performed a molecular assessment of proteostasis pathways in muscles 7 days post-injury (**new Supplementary Fig. 4**). We observed increased markers of autophagy and catabolism in Y2-KO muscles, which also presented with enhanced polyubiquitination (**new Supplementary Fig. 4h-m**). AKT activation was on the other hand similar between groups, and we even observed some increase in mTOR levels, potentially indicating an attempt to outpace increases in protein degradation and compensate for defective growth (**new Supplementary Fig. 4b-g**). Together with the information collected in response to comment #3, we believe this data points at induction of protein catabolism as shared molecular features connecting the failure to fully reach wildtype muscle size during regeneration and the defect in muscle size observed spontaneously over the course of 8 months. Another point worth noting is that the regeneration data indicate that Y2-KO muscles can go back to their baseline size following injury, and they simply re-establish the defect they started with (please see uninjured and day 14 groups in **Fig. 5b**). With this in mind, we believe the defect observed at day 14 post-acute injury is unlikely to be the result of a distinct mechanism when compared to the other models used in the study.

8) The link with ASB2B is of interest but the connection with SMAD2/3 is weak. Just changing the level of expression of Smad3 may be not sufficient to explain the effect on muscle mass. Please show the level of p-Smad2/3 and p-Smad1/5/8 and a ChIP experiment is mandatory to identify whether MuRf1 and Atrogin1 expression is under Smad control in YTHDF2 KO.

We thank the reviewer for suggesting these important experiments. We have now shown an increase p-SMAD3 (**new Fig. 8b-c**); we further show higher levels of cofactor FOXO3a, which has been reported to increase subsequently to SMAD3 activation and to act in concert with SMAD3 in regulating transcription [PMID24920680] (**new Fig. 8b-d**). To determine whether the increased transcriptional activity of MuRF1 and MAFbx were driven by SMAD3, we performed the suggested ChIP experiment and show a significant increase in SMAD3-*Mafbx* binding and a nearly significant enrichment of immunoprecipitated *Murf1* (p=0.0581) in Y2-KO muscles (**new**

Fig. 8g-h), suggesting a contribution of SMAD3 activity to the altered transcriptional profiles. Thank you for helping us to strengthen our proposed mechanism.

Unfortunately, we were unable to detect phosphorylated SMAD2 or 1/5/8 and, also considering that SMAD3 showed the strongest protein level increase (**Supplementary Fig. 6**), have since changed our manuscript text to better focus on SMAD3.

9) Because Smad2/3 has been linked to AKT-FoxO pathway also this signaling must be checked.

As detailed above, we have now shown an increase in total FOXO3a levels with Y2 deficiency (**new Fig. 8b-d**). AKT signaling was on the other hand not changed in our analyses (**new Fig. 4b-d** and **Supplementary Fig. 4b-d**).

10) Finally, rescue experiment in which YTHDF2 is re-expressed in the knockout mice in adulthood and check how much of muscle wasting is reverted is mandatory.

We agree with the reviewer. We now present a recovery study where the tibialis anterior of 2-month-old Y2-KO mice was injected with AAV9-YTHDF2 or control (GFP). Tibialis anterior mass was analyzed 8 weeks following injection and showed rescue of the muscle size phenotype (**new Fig. 1l-m**).

We are grateful to this reviewer for suggesting all of these important experiments that we believe greatly helped to reinforce the study.

Reviewer #3 (Remarks to the Author):

This work reveals the function of m6A reader YTHDF2 in skeletal muscles regulating muscle size. Mechanistically, the author demonstrates that YTHDF2 binds and promotes the decay of m6A modified ASB2 mRNA, which inhibits muscle growth. Knock out of YTHDF2 results in increased ASB2 protein level to further activate the growth inhibition pathway, leading to muscle atrophy in postnatal mice and failure of muscle hypertrophy after overload. The study offers some novel insights on how YTHDF2 functions in muscle growth. I have the following concerns.

Major comments:

1) As a m6A reader, YTHDF2 should have many targets other than ASB2. To fully understand the function of YTHDF2, it is necessary to conduct MeRIP-seq and YTHDF2 RIP-seq or CLIP-seq.

We thank this reviewer for raising this important point. Per their suggestion, we have now added meRIP-Seq and YTHDF2-RIP-Seq and cross-analyzed findings with our proteomics data. First, we performed YTHDF2-RIP-Seq and compared the detected transcripts with those found with m⁶A meRIP-Seq (**new Fig. 6i**); we then compared the overlapping transcripts (136) with the significantly upregulated proteins observed with LC-MS/MS. Of these proteins, we found that 26.7% were differentially expressed in the absence of YTHDF2. Using a cutoff of fold change (FC) > 1.3 for our proteomic results, we observe significant upregulations of eight proteins that correspond to transcripts seen with both meRIP and YTHDF2 RIP: *Asb2*, *Fbxo40*, *Mapkapk3*, *Synpo2l*, *Plin4*, *Cab39*, *Tor1b*, and *Anp32e* (**new Fig. 6j**). When further restricting our cutoff to FC > 2.0, we only observe the upregulation of *ASB2*. While YTHDF2 has variant other targets, our findings nonetheless illustrate a key role for YTHDF2 in modulating *ASB2* expression, and recovery of overloaded Y2-KO muscle mass with si*ASB2* administration (**Fig. 7h-i**) cements a role for the YTHDF2-*ASB2* axis in governing muscle size.

2) In figure 6 and figure 7, why is H9C2 rat myoblast used for the in vitro experiments? This is derived from embryonic BD1X rat heart tissue. The author reports that postnatal heart muscle weight is not affected after YTHDF2 knock out, suggesting YTHDF2 may not regulate ASB2 mRNA decay in heart muscle. Since the animal experiments were performed in mice limb muscles, it is more reasonable to use freshly isolated primary myoblast or at least C2C12 mouse myoblast cells for these experiments.

This is also a good point. With our initial assessments, we performed *Asb2* qPCR on C2C12 and even some non-myocyte cell types, such as NIH3T3, but transcript detection was not clear in those systems, while we saw abundant expression in H9C2s (please see figure below for reviewer only). While H9C2s are isolated from heart tissue, they have not yet fully committed to the cardiac phenotype [PMC4485408], and still share some features with skeletal muscle progenitors [PMID10330239, PMC1155688]. With this rationale, we performed our *in vitro* studies using the H9C2 cell line.

Reviewer Only Figure. qPCR analysis of total *Asb2* mRNA in H9C2, C2C12, and NIH3T3 cells. Comparisons across cell types were analyzed via 1-way ANOVA: ****p≤0.0001. Analyses were conducted using GraphPad Prism Ver 9.4.0 (GraphPad Software).

Minor comments:

1) In page 4, “and histopathological analysis showed no overall disruption of muscle architecture (Fig. k).”, there is no figure K in figure 1.

We apologize for this mistake, which we have now corrected.

We would like to thank this reviewer for their help revising our manuscript and we believe that thanks to their suggestions we now present a strengthened study.

REVIEWER COMMENTS

Reviewer #1 (Remarks to the Author):

The authors addressed all my concerns and questions. I have no additional comments.

Reviewer #2 (Remarks to the Author):

The authors addressed most of my concerns. The paper is improved.

Reviewer #3 (Remarks to the Author):

The authors have answered the two major questions from the first round of review. Regarding the identified YTHDF2 regulated targets from MeRIP and RIP (i.e. Fbxo40, Mapkapk3, Synpo2l, Plin4, Cab39, Tor1b, and Anp32e), the authors should discuss the potential contributions of these targets to the observed phenotype.

Reviewer #1:

The authors addressed all my concerns and questions. I have no additional comments.

We appreciate this reviewer's contribution in strengthening our work.

Reviewer #2:

The authors addressed most of my concerns. The paper is improved.

We thank this reviewer for their help improving the study.

Reviewer #3:

The authors have answered the two major questions from the first round of review. Regarding the identified YTHDF2 regulated targets from MeRIP and RIP (i.e. Fbxo40, Mapkapk3, Synpo2l, Plin4, Cab39, Tor1b, and Anp32e), the authors should discuss the potential contributions of these targets to the observed phenotype.

We thank this reviewer for the suggestion. We have now accordingly edited the discussion section.

REVIEWERS' COMMENTS

Reviewer #3 (Remarks to the Author):

The authors addressed my suggestions. I have no additional comments.